# RA-TBO: Rank-Aware Transfer Bayesian Optimization

## Abstract

Transfer learning accelerates black-box optimization but often falters when source and target tasks exhibit structural distortion or negative correlations. Existing methods that rely on value-based surrogate models struggle to robustly transfer knowledge across such shifts. To overcome these challenges, we propose RA-TBO, a framework that leverages robust ordinal relationships rather than absolute values. By decoupling the process into offline rank-based source modeling and online target calibration, RA-TBO extracts the global structural trend to ensure robust transfer. Furthermore, the framework employs a weighting mechanism based on Kendall's coefficient to exploit latent correlations—whether positive or negative—and adaptively adjust the contribution of source tasks. Comprehensive experiments on synthetic benchmarks and real-world design space exploration problems demonstrate that RA-TBO achieves state-of-the-art performance compared to competitive baselines.

## 1. Introduction

Bayesian Optimization (BO) is widely regarded as the gold standard for expensive black-box problems appearing in hyperparameter tuning (Snoek et al., 2012; Cowen-Rivers et al., 2022), Design Space Exploration (DSE) for processors (Huang et al., 2021; Bai et al., 2021; Zhai & Cai, 2023), chemical synthesis (Burger et al., 2020; Shields et al., 2021), and aerospace design (Lukaczyk et al., 2014; Lam et al., 2018). However, traditional BO suffers from a "cold-start" challenge: in the initial phase, insufficient data leads to inefficient exploration, resulting in wasteful evaluations in unpromising regions and slow convergence. To address this issue, Transfer Bayesian Optimization (TBO) employs transfer learning to accelerate optimization and leverages historical data from source tasks to expedite optimization of the target task.

In real-world applications, source and target tasks often exhibit significant structural distortion and unknown correlations (Jamshidi et al., 2017; Van Aken et al., 2017; Wang et al., 2018). Specifically, in the context of DSE, optimization objectives across different scenarios exhibit pronounced heterogeneity (Bucek et al., 2018). Distinct tasks, such as compute-bound and memory-bound tasks, typically exhibit structural distortion, fundamental discrepancies in the physical meaning and magnitude of evaluation metrics, and a priori unknown positive or negative correlations. Therefore, aligning with the global trend is far more critical and robust than relying on specific numerical values. However, existing methods typically overlook these complexities, constructing value-based surrogate models that neglect the underlying global trend. Moreover, for negatively correlated tasks, these approaches ignore them directly, wasting valuable structural information.

To overcome these limitations, we leverage the philosophy of learning to rank and ordinal optimization. Instead of struggling with fragile numerical regression, we pivot to learning the ordinal structure in source tasks. By focusing on relative rankings, our surrogate model inherently filters out the interference of scale discrepancies and local outliers, offering superior robustness and higher confidence in representing the global trend (Ho et al., 2007). Furthermore, by transferring coarse-grained rank relations and global trends rather than fine-grained, source-specific numerical values, this effectively mitigates the risk of negative transfer under significant distortions. Based on this, we propose RA-TBO, a rank-aware transfer Bayesian optimization framework that decouples the transfer process into an offline phase and an online phase (See Figure 1). First, we construct offline rank-based source models, weighted by Kendall's coefficients, to extract structural priors that leverage both positive and negative correlations. Then, we employ a decoupled architecture that fuses these coarse-grained priors with an online value-based target model to achieve efficient, adaptive optimization.

Our framework is broadly applicable to any black-box optimization with historical data. It consistently matches the performance of non-transfer baselines at a minimum, with-

[1]Anonymous Institution, Anonymous City, Anonymous Region, Anonymous Country. Correspondence to: Anonymous Author <anon.email@domain.com>.

Preliminary work. Under review by the International Conference on Machine Learning (ICML). Do not distribute.

out requiring any prior information. The main contributions are summarized as follows.

1. We propose RA-TBO, a framework that integrates rank-based source surrogates with the value-based target model adaptively via pairwise Kendall's correlations and design a novel rank-aware acquisition function that leverages rank information to guide efficient exploration.

2. We provide theoretical analysis of our framework: concentration bounds for Kendall's rank correlation, the consistency between rank and optimization, and the convergence (no-regret property) of the proposed acquisition function.

3. Extensive experiments on synthetic benchmarks and real-world DSE problems demonstrate that RA-TBO achieves state-of-the-art efficiency and robustness against negative transfer across varying source correlations.

## 2. Related Work

TBO leverages historical data from source tasks to expedite the target task's optimization. According to the components in the BO framework, existing literature can be broadly categorized into surrogate-based, acquisition-based, and initialization methods (Bai et al., 2023).

Surrogate-based methods aim to construct more accurate surrogate models by leveraging information from source tasks. It is also employed in multi-fidelity BO (MFBO), which alternates between the target task and low-fidelity tasks during optimization to reduce costs (Kandasamy et al., 2017; Li et al., 2020; Wu et al., 2020; Sabanza-Gil et al., 2025). TBO distinguishes itself from MFBO by treating source tasks as fixed offline datasets and not interacting with source functions during optimization. Classic methods utilize a covariance matrix for information sharing (Swersky et al., 2013), while other approaches employ Bayesian linear regression with deep neural networks to improve scalability (Snoek et al., 2015; Perrone et al., 2018). Furthermore, recent methods employ few-shot learning and attention mechanisms (Wistuba & Grabocka, 2021; Wei et al., 2021). To mitigate negative transfer on different tasks, some methods train independent source models and fuse them online. RGPE (Feurer et al., 2018) weights source models based on rank loss, while SGPR (Golovin et al., 2017) employs residual learning for modeling. TransBO (Li et al., 2022) proposes a two-phase framework that solves a constrained optimization problem with ranking loss to balance the weights of source and target surrogates.

Some studies alternatively consider transferring knowledge through the acquisition function to guide the target search. Representative approaches include Multi-task BO (Swersky et al., 2013; Wu & Frazier, 2016; Moss et al., 2020), Ensemble GP-based (Wistuba et al., 2018), and Reinforcement

Learning-based acquisition function (Volpp et al., 2020; Hsieh et al., 2021).Unlike guiding the search process, initialization methods aim to identify high-quality initial points to accelerate optimization (Kim et al., 2017; Wistuba & Grabocka, 2021; Wei et al., 2021; Chen et al., 2022).

Our framework integrates surrogate-based and acquisition-based paradigms. Distinct from existing works that rely on value-based models in transfer learning, we decouple the modeling process by employing heterogeneous loss functions for source and target tasks, which are subsequently fused into an ensemble surrogate model. Furthermore, we inject rank information into the acquisition function to accelerate convergence.

We present RA-TBO, a rank-aware Bayesian Optimization framework that overcomes structural distortion and scale discrepancies and exploits previously unknown correlations—both positive and negative.

## 3. Problem Formulation and Background

In this section, we define the problem of Transfer Learning with multiple source tasks, and review the preliminaries of BO and ranking loss functions. For clarity, a comprehensive table of notations is provided in Appendix A.

We address the sequential global optimization of an expensive black-box function $f : \mathcal{X} \to \mathbb{R}$ over a finite horizon $T$. At iteration $t$, the available information consists of the *accumulated* target observations $\mathcal{D}_t = \{(\mathbf{x}_i, y_i)\}_{i=1}^{n_t}$ and *static* historical observations $\mathcal{D}_k = \{(\mathbf{x}_i^k, y_i^k)\}_{i=1}^{n_k}$ from $K$ source tasks $f_k : \mathcal{X} \to \mathbb{R}$ (where $n_k \gg n_t, k = 1, \cdots, K$). Our goal is to accelerate the optimization process by leveraging the shared structure inherent in these source tasks.

**Bayesian Optimization (BO)** provides an efficient framework for expensive black-box optimization by iteratively selecting the next query point $x_{t+1} = \arg\max \alpha(x|\mathcal{D}_t)$ guided by probabilistic surrogate models and acquisition functions. The Gaussian Process (GP) is the standard surrogate model, which assumes a joint Gaussian prior. By conditioning on $\mathcal{D}_t$, the posterior at a query point $x$ is analytically derived as $\mathcal{N}(\mu_t(x), \sigma_t^2(x))$ with moments: $\mu_t(x) = \mathbf{k}_t(x)^\top (\mathbf{K}_t + \sigma_n^2 \mathbf{I})^{-1} \mathbf{y}_t, \sigma_t^2(x) = k(x, x) - \mathbf{k}_t(x)^\top (\mathbf{K}_t + \sigma_n^2 \mathbf{I})^{-1} \mathbf{k}_t(x)$, where $\mathbf{K}_t = [k(x_i, x_j)]_{i,j=1}^t$ and $\mathbf{k}_t(x) = [k(x_i, x)]_{i=1}^t$, and $\sigma_n^2$ represents the observation noise variance. For large-scale datasets, the $\mathcal{O}(n^3)$ computational complexity of GPs renders training and inference prohibitive. To address this, Deep Ensembles $\{h_t^{(r)}(\mathbf{x})\}_{r=1}^R$ are applied to serve as a scalable alternative to GP (Lakshminarayanan et al., 2017). We focus on epistemic uncertainty, as it is the primary driver of exploration in BO. The predictive moments are estimated as: $\mu_t(\mathbf{x}) = \frac{1}{R} \sum_{r=1}^R h_t^{(r)}(\mathbf{x}), \sigma_t^2(\mathbf{x}) =$

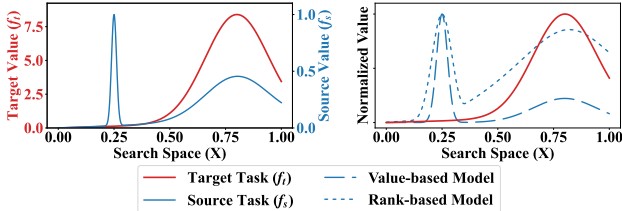

*Figure 1.* RA-TBO framework workflow. The workflow consists of an Offline Phase for training models on source datasets, and an Online Phase for the target task. By quantifying the similarity between source and target tasks, the method fuses source knowledge with the target GP to construct the final surrogate model. It employs Rank-aware UCB to select the next query.

$$\frac{1}{R-1}\sum_{r=1}^{R}\left(h_t^{(r)}(\mathbf{x})-\mu_t(\mathbf{x})\right)^2 \quad \text{and} \quad \text{Cov}_t(\mathbf{x},\mathbf{x}') =$$
$$\frac{1}{M-1}\sum_{m=1}^{M}\left(h_t^{(m)}(\mathbf{x})-\mu_t(\mathbf{x})\right)\left(h_t^{(m)}(\mathbf{x}')-\mu_t(\mathbf{x}')\right).$$

Finally, the acquisition function $\alpha(\mathbf{x}|\mathcal{D}_t)$ guides the search by trading off exploration and exploitation. Common choices include Probability of Improvement (PI), Expected Improvement (EI), and Upper Confidence Bound (UCB) (Shahriari et al., 2015; Frazier, 2018).

**Rank Loss functions** are employed to train the ordinal surrogate models. Two standard objectives are used in this work: the Pairwise approach (Qin et al., 2008) and the Listwise approach (Cao et al., 2007), which have been proven effective in offline model-based optimization (Tan et al., 2025). Given the accurate scores $\mathbf{y} \in \mathbb{R}^n$ and the predicted scores $\hat{\mathbf{y}} \in \mathbb{R}^n$, the Pairwise approach focuses on relative rank using cosine similarity, formulated as $\mathcal{L}_{pair}(\mathbf{y},\hat{\mathbf{y}}) = 1 - \frac{(\mathbf{y}-\bar{y})^\top(\hat{\mathbf{y}}-\bar{\hat{y}})}{\|\mathbf{y}-\bar{y}\|_2\|\hat{\mathbf{y}}-\bar{\hat{y}}\|_2}$, where $\bar{y}$ and $\bar{\hat{y}}$ denote the arithmetic means of the score vectors $\mathbf{y}$ and $\hat{\mathbf{y}}$. While the Listwise approach emphasizes the global distribution using cross-entropy, defined as $\mathcal{L}_{list}(\mathbf{y},\hat{\mathbf{y}}) = -\sum_{i=1}^n \frac{\exp(y_i)}{\sum_{j=1}^n \exp(y_j)} \log\left(\frac{\exp(\hat{y}_i)}{\sum_{j=1}^n \exp(\hat{y}_j)}\right)$.

# 4. Rank-Aware Transfer Bayesian Optimization

This section details the proposed framework as illustrated in Figure 1, covering the motivation, similarity quantification, surrogate modeling, and acquisition function formulation.

*Figure 2.* Motivation for Rank-based Modeling. **Left**: a source task with a misleading local optimum and scale discrepancies compared to the target. **Right**: the normalized outputs of the value-based, rank-based model and the target task.

## 4.1. Motivation

In transfer learning settings, source tasks often exhibit structural distortion and scale discrepancies relative to the target task, making it crucial to leverage the structural trend rather than the values. However, existing methods construct source surrogate models via Mean Squared Error (MSE). This excessive focus on fitting exact numerical values leads to a neglect of the underlying structure. Inspired by recent advances in offline model-based optimization, where rank-based models are employed to reduce out-of-distribution errors (Tan et al., 2025), we observe that in source tasks, it is not necessary to predict true values precisely, but rather to predict relative rankings. As shown in Figure 2, although the source and target tasks share similar global structures, they suffer from scale discrepancies and local shifts. The source task presents a misleading local optimum at $x = 0.25$ and overshadows the true global optimal structure shared with the target task. The value-based model, which aims to fit absolute numerical values, remains dominated by this local spike in the source task, even after normalization. In contrast, the rank-based model aims to capture the global structure across tasks via ordinal relationships. This mitigates interference from scale discrepancies and local outliers, thereby providing a robust structural prior for the target task.

Although ranking models incur information loss compared to value-based models, in source tasks, given the abundance of data and the bias introduced by the target task, capturing relative rankings is sufficient and more robust. For the target task, we retain value-based models to preserve this expensive information.

## 4.2. Task Similarity Quantification

Given that source tasks exhibit prior unknown correlations with the target task, indiscriminately transferring all source tasks equally often results in negative transfer. To dynamically quantify the structural similarity between source and target tasks, we use Rank-Aware Kendall's $\tau$.

Firstly, we employ deep ensembles to construct value-based surrogate models $V_k(\mathbf{x})$ for source tasks $f_k$ ($k = 1, \cdots, K$) using large-scale datasets. This leverages abundant offline data to ensure that $V_k$ approximates the ground truth $f_k$ with high fidelity. Consequently, we substitute $f_k$ with $V_k$ in our theoretical framework, and treat the correlation between $f_k$ and $f$ as effectively represented by that between $V_k$ and $f$. The value-based models $V_k$ are solely for similarity quantification to avert premature loss of distributional details. We then project this preserved distributional information into the ranking space using ranking metrics to ensure robust transfer. Thus, while we leverage the value information in source datasets, the framework's intrinsic nature remains strictly rank-based.

Then at iteration $t$, we measure the ranking capability of each $V_k$ by applying it to predict the relative order of the target task observations $\mathcal{D}_t = \{(\mathbf{x}_i, y_i)\}_{i=1}^{n_t}$. For any pair of query points $\mathbf{x}_i, \mathbf{x}_j \in \mathcal{D}_t$, we define the probability that the $k$-th source surrogate model $V_k$ ranks $\mathbf{x}_i$ higher than $\mathbf{x}_j$ as:

$$
\begin{aligned}
p_{ij}^k &= \mathbb{P}(V_k(\mathbf{x}_i) > V_k(\mathbf{x}_j)) \\
&= \Phi\left( \frac{\mu_{\hat{k}}(\mathbf{x}_i) - \mu_{\hat{k}}(\mathbf{x}_j)}{\sqrt{\sigma_{\hat{k}}^2(\mathbf{x}_i) + \sigma_{\hat{k}}^2(\mathbf{x}_j) - 2\mathrm{Cov}_{\hat{k}}(\mathbf{x}_i, \mathbf{x}_j)}} \right),
\end{aligned} \quad (1)
$$

where $V_k(\mathbf{x}) \sim \mathcal{N}(\mu_{\hat{k}}(\mathbf{x}), \sigma_{\hat{k}}^2(\mathbf{x}))$, $\mathrm{Cov}_{\hat{k}}$ denotes the empirical covariance and $\Phi(\cdot)$ is the cumulative distribution function (CDF) of the standard normal distribution. By assessing whether the model predictions accord with the ground truth, we calculate the expected number of concordant and $N_d^k$ discordant pairs $N_c^k$ for the $k$-th source task:

$$
\begin{aligned}
\mathbb{E}[N_c^k] &= \sum_{1 \le i < j \le n_t} \left[ \mathbb{I}(y_i > y_j) p_{ij}^k + \mathbb{I}(y_i < y_j)(1 - p_{ij}^k) \right] \\
\mathbb{E}[N_d^k] &= \sum_{1 \le i < j \le n_t} \left[ \mathbb{I}(y_i > y_j)(1 - p_{ij}^k) + \mathbb{I}(y_i < y_j) p_{ij}^k \right],
\end{aligned}
$$
$$(2)$$

where $\mathbb{I}(\cdot)$ denotes the indicator function, and the Expected Kendall's Rank Correlation Coefficient can be defined:

$$
\tau_k = \frac{\mathbb{E}[N_c^k] - \mathbb{E}[N_d^k]}{\binom{n_t}{2}} \in [-1, 1]. \quad (3)
$$

Notably, in a pairwise ranking framework, a negative correlation also implies an inverted structural relationship, and it is as informative as a positive correlation for transfer learning. So we quantify the similarity weight $w_k$ for the $k$-th source task as:

$$
w_k = |\tau_k| \in [0, 1]. \quad (4)
$$

As $w_k \to 1$, we leverage more information from the source task during modeling. Furthermore, as the optimization

proceeds and high-performance points accumulate in $\mathcal{D}_t$, the similarity pays more attention to high-performance regions, which aligns with the maximization objective.

### 4.3. Surrogate Model

Having quantified the similarity, the next step is to fuse the rank-based model $\{F_k\}_{k=1}^K$ for the source tasks with the value-based surrogate model $F$ for the target task. Given a single query point $\mathbf{x}$, $F(\mathbf{x}) \sim \mathcal{N}(\mu(\mathbf{x}), \sigma^2(\mathbf{x}))$ and $F_k(\mathbf{x}) \sim \mathcal{N}(\mu_k(\mathbf{x}), \sigma_k^2(\mathbf{x}))$. Here, we employ a rank-based loss function and Deep Ensembles (serving as a scalable approximation of GP) to construct $\{F_k\}_{k=1}^K$ for the large-scale $\mathcal{D}_k$, and utilize a standard GP to construct the value-based $F$ for the limited $\mathcal{D}_t$.

**Model Normalization.** Since $\{F_k\}_{k=1}^K$ are rank-based models, their predictive values lack physical meaning and exhibit significant scale discrepancies with $F$. To address this, we employ a Normalization set:

$$
\mathcal{X}_{norm} = \{\mathbf{x} \mid (\mathbf{x}, y) \in \mathcal{D}_t\} \cup \bigcup_{k=1}^K \{\mathbf{x} \mid (\mathbf{x}, y) \in \mathcal{D}_k\}. \quad (5)
$$

and calculate the calibration statistics (mean $\mu_{norm}$ and standard deviation $\sigma_{norm}$) based on the predictions of $\{F_k\}_{k=1}^K$ and $F$ over $\mathcal{X}_{norm}$. To use both positive and negative information, we rectify the prediction direction using the sign of $\tau_k$ in Equation (3). Consequently, the normalized predictive mean $\tilde{\mu}$ and standard deviation $\tilde{\sigma}$ are obtained as:

$$
\begin{aligned}
\tilde{\mu}(\mathbf{x}) &= \frac{\mu(\mathbf{x}) - \mu_{\mathrm{norm}}^F}{\sigma_{\mathrm{norm}}^F}, \tilde{\sigma}(\mathbf{x}) = \frac{\sigma(\mathbf{x})}{\sigma_{\mathrm{norm}}^F}, \\
\tilde{\mu}_k(\mathbf{x}) &= \mathrm{sgn}(\tau_k) \frac{\mu_k(\mathbf{x}) - \mu_{\mathrm{norm}}^k}{\sigma_{\mathrm{norm}}^k}, \tilde{\sigma}_k(\mathbf{x}) = \frac{\sigma_k(\mathbf{x})}{\sigma_{\mathrm{norm}}^k},
\end{aligned} \quad (6)
$$

where $\mathrm{sgn}(\cdot)$ denotes the sign function. When the size of $\mathcal{X}_{norm}$ is excessive, a random subset of $\mathcal{X}_{norm}$ can be employed for computational efficiency.

**Probabilistic Fusion.** Given a finite set of query points $\mathbf{X}$, assuming the normalized target model $\tilde{F}(\mathbf{X}) \sim \mathcal{N}(\tilde{\mu}, \tilde{\Sigma})$ and source models $\tilde{F}_k(\mathbf{X}) \sim \mathcal{N}(\tilde{\mu}_k, \tilde{\Sigma}_k)$ independently predict the underlying objective, we employ the Generalized Product of Experts (gPoE) framework (Deisenroth & Ng, 2015) to fuse them. The fused distribution $p_{fuse}(\mathbf{y}|\mathbf{X})$ is defined as the product of the target belief (with weight 1) and the beliefs of source experts $\tilde{F}_k$ weighted by $w_k$:

$$
p_{fuse}(\mathbf{y}|\mathbf{X}) \propto p_{\tilde{F}}(\mathbf{y}|\mathbf{X}) \prod_{k=1}^K \left[ p_{\tilde{F}_k}(\mathbf{y}|\mathbf{X}) \right]^{w_k}, \quad (7)
$$

where $\mathbf{y}$ denotes the vector of predicted values. As $w_k \to 0$, the contribution of the corresponding source expert vanishes and reduces the risk of negative transfer from unrelated tasks.

As derived in Appendix B, the fused surrogate model retains the Gaussian Process distribution property: $F_{fuse}(\mathbf{X}) \sim \mathcal{GP}(\boldsymbol{\mu}_{fuse}, \boldsymbol{\Sigma}_{fuse})$, where:

$$\boldsymbol{\Sigma}_{fuse} = \left( \tilde{\boldsymbol{\Sigma}}^{-1} + \sum_{k=1}^{K} w_k \tilde{\boldsymbol{\Sigma}}_k^{-1} \right)^{-1},$$

$$\boldsymbol{\mu}_{fuse} = \boldsymbol{\Sigma}_{fuse} \left( \tilde{\boldsymbol{\Sigma}}^{-1} \tilde{\boldsymbol{\mu}} + \sum_{k=1}^{K} w_k \tilde{\boldsymbol{\Sigma}}_k^{-1} \tilde{\boldsymbol{\mu}}_k \right). \quad (8)$$

For a single query point $\mathbf{x}$, the fused surrogate model follows a univariate Gaussian distribution $F_{fuse}(\mathbf{x}) \sim \mathcal{N}(\mu_{fuse}(\mathbf{x}), \sigma_{fuse}^2(\mathbf{x}))$.

This fusion strategy establishes a dynamic trust mechanism: as the target task optimization progresses and its uncertainty diminishes, the fused model automatically places more reliance on the target model $F$. Meanwhile, the iterative retraining requires only the limited target data $\mathcal{D}_t$ and circumvents the burden of the large-scale source data $\{\mathcal{D}_k\}_{k=1}^{K}$, making it computationally efficient.

### 4.4. Acquisition Function

To fully leverage the structure information from source tasks, we use Rank-Aware Upper Confidence Bound (RA-UCB) as the Acquisition Function:

$$\alpha_t(\mathbf{x}) = \underbrace{\mu_{fuse}(\mathbf{x}) + \sqrt{\beta_t} \sigma_{fuse}(\mathbf{x})}_{\text{Standard UCB}} + \underbrace{\lambda_t \cdot \mathrm{REN}(\mathbf{x}; \mathcal{A}_{src})}_{\text{Rank Entropy}}, \quad (9)$$

where $\beta_t$ and $\lambda_t$ are hyperparameters. The Rank Entropy (REN) term quantifies the uncertainty of the ordinal relationship between the candidate $\mathbf{x}$ and the set of high-quality anchor points $\mathcal{A}_{src}$ derived from source datasets, which is constructed by taking the union of the top-$M$ outcomes selected individually from each source task.

$$\mathcal{A}_{src} = \bigcup_{k=1}^{K} \{\mathbf{x} \mid (\mathbf{x}, y) \in \mathcal{D}_k, y \text{ is in top-}M \text{ of } \mathcal{D}_k\}. \quad (10)$$

Then the REN is calculated as the average binary entropy:

$$\mathrm{REN}(\mathbf{x}; \mathcal{A}_{src}) = \frac{1}{|\mathcal{A}_{src}|} \sum_{\mathbf{a} \in \mathcal{A}_{src}} H(p_{\mathbf{x},\mathbf{a}}), \quad (11)$$

where $H(p) \approx p(1-p)$ serves as a quadratic approximation of the binary entropy and $p_{\mathbf{x},\mathbf{a}}$ represents the probability predicted by the fused model that $\mathbf{x}$ outperforms $\mathbf{a}$:

$$p_{\mathbf{x},\mathbf{a}} = \Phi \left( \frac{\mu_{fuse}(\mathbf{x}) - \mu_{fuse}(\mathbf{a})}{\sqrt{\sigma_{fuse}^2(\mathbf{x}) + \sigma_{fuse}^2(\mathbf{a}) - 2\mathrm{Cov}_{fuse}(\mathbf{x},\mathbf{a})}} \right). \quad (12)$$

---

**Algorithm 1** The RA-TBO Framework

---

**Input:** Target task $f$, Source datasets $\{\mathcal{D}_k\}_{k=1}^{K}$, Initial target data $\mathcal{D}_{init}$, Optimization iterations $T$, $\beta_t, \lambda_t, M$.
**Output:** Best found solution $\mathbf{x}^*$.
**Offline Phase:**
**for** $k = 1$ **to** $K$ **do**
    Construct value-based models $V_k$ and rank-based models $F_k$ on $\mathcal{D}_k$ via Deep Ensembles.
**end for**
Construct $\mathcal{A}_{src}$ in Equation (10)
**Online Phase:**
Initialize $\mathcal{D}_0 \leftarrow \mathcal{D}_{init}$.
**for** $t = 1$ **to** $T$ **do**
    Train value-based target GP model $F$ on $\mathcal{D}_t$.
    **for** $k = 1$ **to** $K$ **do**
        Compute $\tau_k$ in Equation (3) and set $w_k = |\tau_k|$.
    **end for**
    Normalize $F$ and $F_k$ in Equation (6)
    Construct the fused model $F_{fuse}$ in Equation (8)
    Find $\mathbf{x}_{new} = \arg\max_{\mathbf{x}} \alpha_t(\mathbf{x})$ in Equation (9):
    Evaluate $y_{new} = f(\mathbf{x}_{new})$.
    $\mathcal{D}_t \leftarrow \mathcal{D}_{t-1} \cup \{(\mathbf{x}_{new}, y_{new})\}$.
**end for**
**Return:** Best found solution $\mathbf{x}^*$ in $\mathcal{D}_T$.

---

The entropy term $H(p)$ peaks when $p_{\mathbf{x},\mathbf{a}} = 0.5$, which means the model cannot distinguish the relative order between $\mathbf{x}$ and $\mathbf{a}$. This explicitly encourages exploration in these ambiguous regions to resolve ranking uncertainty. Furthermore, this term is differentiable with respect to $\mathbf{x}$ and can be efficiently maximized via gradient-based methods.

To ensure computational efficiency, we compute the covariance term by jointly inputting the candidate $\mathbf{x}$ and the anchor set $\mathcal{A}_{src}$ exclusively for the REN calculation. In contrast, for the standard UCB term, which depends solely on pointwise uncertainty, we compute the marginal variance for $\mathbf{x}$ individually. Algorithm 1 outlines the complete workflow of our proposed framework.

## 5. Theoretical Analysis

In this section, we provide theoretical guarantees for the RA-TBO framework, which focus on concentration bounds for Kendall's rank correlation coefficient, consistency between ranking and optimization, and regret bounds and convergence properties of our acquisition function.

### 5.1. Concentration Bounds of Rank Correlation

As we estimate the actual population correlation $\tau_k$ based on limited target observations, we analyze the empirical estimator's concentration properties via Hoeffding's inequality

(Hoeffding, 1963) and a variance-adaptive bound via Bernstein's inequality.

**Theorem 5.1** (Concentration of Correlation Estimation). *Assuming $\hat{\tau}_k$ is the empirical estimator from $n$ independent and identically distributed target observations, and $\tau_k$ is the true value. For any precision $\epsilon > 0$:*

1. ***General Bound:*** $\mathbb{P}(|\hat{\tau}_k - \tau_k| \geq \epsilon) \leq 2\exp(-\frac{n\epsilon^2}{4})$.

2. ***Adaptive Bound:*** *There exists a constant $C$ such that* $\mathbb{P}(|\hat{\tau}_k - \tau_k| \geq \epsilon) \leq 2\exp(-\frac{n\epsilon^2}{8(1-\tau_k^2)+C\epsilon})$.

Both bounds in Theorem 5.1 demonstrate that the estimation error converges exponentially as the sample size $n$ increases. Furthermore, the second inequality proves that the convergence rate accelerates from $\mathcal{O}(e^{-n\epsilon^2})$ to $\mathcal{O}(e^{-n\epsilon})$ as $|\tau_k| \to 1$. The complete analysis is in Appendix C.

### 5.2. Optimization Consistency

To explain the rationale of exploiting ordinal relationships for optimization, we establish the consistency between rank correlation and the location of the global optimum. Unlike prior analyses that rely on complete order preservation (Tan et al., 2025; Lv et al., 2024), we construct our proof from the relationship between Kendall's correlation coefficient and the positional error of the optimal solution.

**Theorem 5.2** (Optimization Consistency). *Assume the target function $f$ and the $k$-th source surrogate $f_k$ are defined on a compact domain $\mathcal{X} \subset \mathbb{R}^d$ and are Lipschitz continuous. Assume $f$ satisfies a local growth condition of order $\alpha \geq 1$ near its unique global optimum $\mathbf{x}^*$, and $f_k$ satisfies a local growth condition of order $\alpha_k \geq 1$ near its optimum $\mathbf{x}_k^*$. Then, when $\tau_k \to 1$, there exist constants $C_1 > 0$ and $C_2 > 0$ (depending on the geometric properties) such that the positional error is bounded by:*

$$\|\mathbf{x}_k^* - \mathbf{x}^*\| \leq C_1 \cdot (1 - \tau_k)^{\frac{1}{dC_2}}. \tag{13}$$

Theorem 5.2 demonstrates that transferring and learning from ranks is consistent with optimization. These assumptions are commonly satisfied in real-world black-box optimization. Although these conditions are necessary for the strict theoretical bounds, our framework demonstrates empirical effectiveness in complex real-world applications, where these conditions may be relaxed or unverifiable. The detailed proof and explanation are provided in Appendix D.

### 5.3. Regret Bound and Convergence

As the fused surrogate model in Equation (8) is also a Gaussian Process, we follow the theoretical framework established in (Srinivas et al., 2010) and extend the cumulative regret from the standard UCB to RA-UCB. Although the

REN term introduces an additional objective, we prove that the algorithm remains no-regret if $\lambda_t$ decays appropriately.

**Theorem 5.3** (No-Regret Property). *Assume the standard GP-UCB parameter $\beta_t$ ensures high-probability coverage. With the rank exploration weight decaying as $\lambda_t = \mathcal{O}(t^{-1/2})$, the algorithm achieves vanishing average regret:*

$$\lim_{T \to \infty} \frac{R_T}{T} = 0. \tag{14}$$

Theorem 5.3 demonstrates that the algorithm preserves the no-regret property despite the addition of the REN term. Although this term introduces a bounded theoretical cost, our ablation experiments show that it effectively accelerates convergence to the global optimum. The detailed analysis is in Appendix E.

## 6. Experiments

In this section, to comprehensively evaluate the RA-TBO framework, we validate its optimization efficiency on both synthetic benchmarks and real-world DSE problems, its robustness against source tasks of varying similarity, and the necessity of each module. Additionally, a detailed analysis of the ablation experiments, computational complexity, and sensitivity analysis is provided in Appendix F.3.

### 6.1. Experimental Setup

**Benchmarks.** We evaluate our framework on a diverse suite of benchmarks, which ranges from synthetic functions to real-world DSE problems.

*Synthetic Benchmarks:* We employ five widely used test functions as our target task: Ackley, Branin, Hartmann3, Hartmann6, and Schwefel. All synthetic benchmarks are formulated as maximization problems. For the Ackley function, we conduct tests at input dimensions $d = 2$ and $d = 50$ to verify the algorithm's scalability to high-dimensional settings. As for source tasks, we apply scaling and shifting transformations to these functions. Detailed information is provided in Appendix F.1.

*Real-world DSE Problems:* We evaluate our framework on DSE problems that aim to optimize processor parameter configurations with a limited evaluation budget across different benchmarks. Based on experts' knowledge, we select 50 important register-level and software-level parameters for optimization. We employ two classical test suites: (1) **Renaissance:** A benchmark where the objective is to **minimize** the completion time; (2) **emu-MySQL:** A benchmark where the objective is to **maximize** query per second (QPS). This benchmark consists of three sub-benchmarks: Readonly, Writeonly, and Default, which can be evaluated independently. Based on them, we design five transfer scenarios. For negative correlation settings, Renaissance and

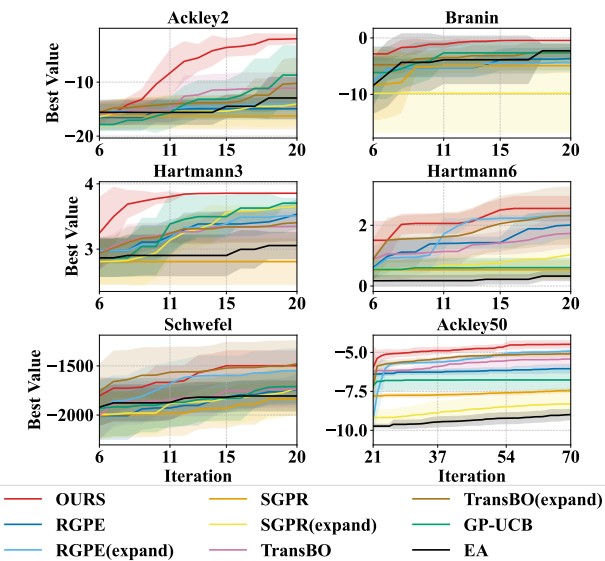

*Figure 3.* Optimization curves for Synthetic Benchmarks.

emu-MySQL serve as the source and target tasks, respectively, driven by their conflicting optimization objectives. For positive correlation settings, we run sub-benchmark experiments in emu-MySQL, selecting one sub-benchmark as the target and the other two as sources. Notably, all these correlations serve only as qualitative indicators derived from expert knowledge, as an exhaustive evaluation of the entire design space is infeasible. We feed the raw objective values into all methods without correlation information.

**Methods.** To validate the optimization efficiency of RA-TBO, we compare it against five representative baselines, categorized into non-transfer and transfer learning methods. For non-transfer methods, we employ Standard BO with GP as the representative model-based optimization approach, and an Evolutionary Algorithm (EA) as the model-free baseline. For transfer learning methods, we use three advanced methods, including RGPE (Feurer et al., 2018), SGPR (Golovin et al., 2017), and TransBO (Li et al., 2022). These transfer methods typically assume positive correlations. To demonstrate that the superiority of RA-TBO stems from its intrinsic design rather than a simple sign-flipping heuristic, we also expand the source tasks for RGPE, SGPR, and TransBO by including both the original source datasets $\{\mathcal{D}_k\}_{k=1}^K$ and their sign-flipped datasets $\{(\mathbf{x}, -y) \mid (\mathbf{x}, y) \in \mathcal{D}_k\}_{k=1}^K$. For all these compared methods, we use the OpenBox toolkit (Li et al., 2021; Jiang et al., 2024), an open-source system designed for efficient and generalized black-box optimization. We adopt the default configurations provided by the system, which are empirically optimized for superior robustness. Consistent with our RA-UCB acquisition function in Equation (9), we employ

UCB for all comparative methods, using the same $\beta_t$ as ours. We utilize the Listwise approach as the ranking loss function in our framework. Additional detailed configurations are provided in Appendix F.2. In the initialization phase, we employ Latin Hypercube Sampling (LHS) to generate the initial set of observations. For low-dimensional synthetic benchmarks, we set $N_{init} = 5$ and $T = 15$. In these cases, we utilize $K = 2$ source tasks, each containing $100 \times d$ historical observations. In contrast, for the high-dimensional Ackley ($d = 50$) and the real-world application, we increase the budget to $N_{init} = 20$ and $T = 50$. Regarding the source tasks, we use $K = 2$ for Ackley ($d = 50$) and the positive real-world settings, while employing $K = 1$ for the inverse real-world settings; each source task comprises 1,000 historical data points. To ensure statistical reliability, all experiments are repeated five times with different random seeds. All methods share the same initial observations and source task datasets.

### 6.2. Optimization Performance

**Results on Synthetic Benchmarks.** Figure 3 presents the optimization curves for the synthetic benchmarks. The solid curves indicate the mean performance, and the shaded regions represent $\pm 1$ standard deviation over 5 independent optimization trials. As evidenced by the results, even with the expanded source tasks provided to the transfer baselines, RA-TBO achieves the highest objective value and the fastest convergence rate in other benchmarks. A striking observation is that, in some benchmarks, competitive transfer learning baselines perform even worse than the non-transfer baseline, indicating that they suffer from **negative transfer**, despite their source datasets being expanded. This phenomenon is mainly caused by nonlinear shifts and scale transformations applied to source tasks, which place greater demands on the algorithm's robustness in leveraging them. By focusing on relative rankings rather than absolute values, our framework demonstrates superior ability to leverage source tasks when other methods fail.

**Results on DSE problems.** Table 1 presents the normalized performance improvements of different methods compared to the default configuration on DSE problems. As demonstrated by the results, RA-TBO outperforms all baselines across both positive and negative correlation scenarios. This highlights its practical value in real-world applications where the correlations between source and target tasks are entirely unknown and the metrics are heterogeneous.

### 6.3. Robustness and Ablation

**Robustness.** To evaluate the robustness of RA-TBO to varying degrees of task similarity, we constructed source tasks with a range of Kendall rank correlation coefficients on synthetic benchmarks and compared their optimization

*Table 1.* Performance Improvement (%) on DSE problems. Results are reported as mean ± std.

| METHOD | RENAISSANCE | MYSQL | READONLY | WRITEONLY | DEFAULT |
|---|---|---|---|---|---|
| GP-UCB | 6.07 ± 0.47 | 2.97 ± 0.29 | 3.26 ± 0.30 | 3.77 ± 0.31 | 3.56 ± 0.26 |
| EA | 3.15 ± 0.45 | 1.67 ± 0.25 | 1.63 ± 0.22 | 1.62 ± 0.26 | 1.70 ± 0.31 |
| RGPE | 5.23 ± 0.37 | 3.03 ± 0.17 | 3.73 ± 0.36 | 3.88 ± 0.28 | 3.98 ± 0.33 |
| SGPR | 5.47 ± 0.52 | 2.32 ± 0.13 | 4.02 ± 0.35 | 3.82 ± 0.31 | 3.57 ± 0.41 |
| TRANSBO | 6.02 ± 0.32 | 3.17 ± 0.22 | 3.97 ± 0.40 | 4.46 ± 0.36 | 3.85 ± 0.43 |
| RGPE(EXPAND) | 6.42 ± 0.31 | 3.04 ± 0.26 | 3.21 ± 0.51 | 3.89 ± 0.40 | 3.74 ± 0.27 |
| SGPR(EXPAND) | 5.92 ± 0.27 | 2.90 ± 0.33 | 3.87 ± 0.46 | 3.78 ± 0.39 | 3.67 ± 0.42 |
| TRANSBO(EXPAND) | 7.12 ± 0.33 | 3.19 ± 0.29 | 3.77 ± 0.34 | 4.42 ± 0.41 | 3.91 ± 0.37 |
| **RA-TBO** | **8.29 ± 0.23** | **3.76 ± 0.32** | **4.23 ± 0.42** | **4.53 ± 0.49** | **4.33 ± 0.39** |

curves. These correlations range from $\tau = -1.0$ to $\tau = 1.0$, achieved through controlled scaling and shifting transformations applied to the target task.

As illustrated in Figure 4, the optimization curves underscore the dynamic trust mechanism of RA-TBO, which adaptively quantifies similarity to modulate the influence of source information. Benefiting from the pairwise ranking formulation of our Kendall similarity, RA-TBO effectively leverages both positive ($\tau = 1.0$) and inverse ($\tau = -1.0$) correlations to accelerate convergence. For medium correlations ($\tau = \pm 0.5$), our framework extracts partial structural consistency to yield moderate gains. Crucially, when the source task is irrelevant to the target task ($\tau = 0$), the algorithm progressively downweights the source's contribution during optimization. This mechanism allows the model to filter out irrelevant information gradually and ensures that the final performance remains on par with the non-transfer baseline. This demonstrates that our framework successfully quantifies source task similarity and assigns weights accordingly, effectively preventing the model from being misled by unrelated information.

**Ablation Experiments.** To verify the individual contributions of the rank-based surrogate, we conducted ablation experiments on the rank-based model for source tasks to transfer, the similarity quantification with the value-based model, the Listwise rank loss function, and the Rank Entropy (REN) term. The detailed experimental settings and results are provided in Appendix F.3. In summary, our analysis confirms that the full RA-TBO framework consistently outperforms all ablated variants. Specifically, the results validate that: (1) rank-based modeling offers superior robustness under distribution shifts compared to value-based surrogates; (2) using a value-based model combined with Kendall's $\tau$ for similarity estimation not only preserves theoretical analyzability, but also averts the premature loss of distributional details; (3) Listwise loss preserves global structural information more effectively than pairwise methods, and (4) the REN term is essential for promoting efficient exploration. The seamless integration of these complementary components

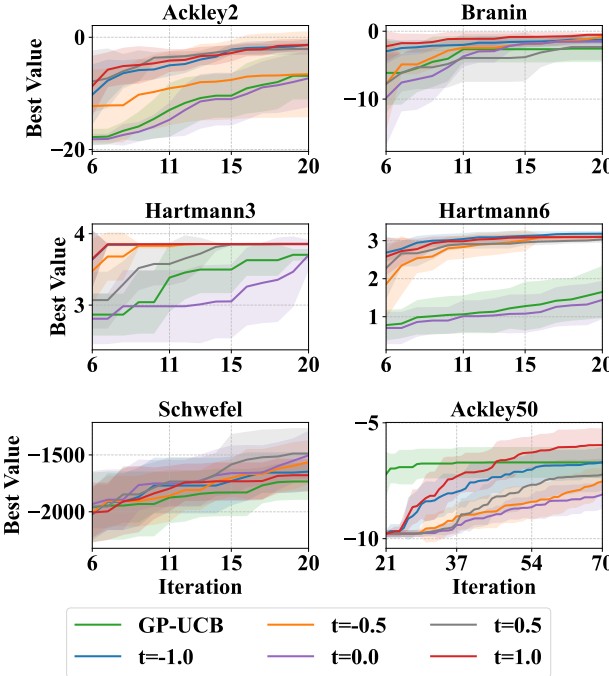

*Figure 4.* Optimization curves for the robustness experiments.

enables RA-TBO to achieve optimal performance.

## 7. Conclusion

In this paper, we present RA-TBO, a Transfer Bayesian Optimization framework to overcome the limitations of traditional value-based methods in handling nonlinear shifts and negative task correlations. By decoupling the transfer process into offline rank-based source modeling and online value-based target modeling, RA-TBO effectively extracts structural invariants to achieve robust transfer. Experimental results on both numerical benchmarks and real-world DSE problems demonstrate the superiority of RA-TBO over state-of-the-art methods.

## Impact Statement

This paper presents work whose goal is to advance the field of Machine Learning. There are many potential societal consequences of our work, none of which we feel must be specifically highlighted here.

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

## A. Table of Notations

In this section, we summarize the key notations used throughout the paper in Table 2.

*Table 2.* Summary of Notations

| Notation | Description |
|---|---|
| **1. Problem Formulation & Data** | |
| $f(\cdot)$ | The target task function, $f : \mathcal{X} \to \mathbb{R}$. |
| $\mathcal{X}$ | The shared search space |
| $d$ | The dimension of the shared search space ($\mathcal{X} \subseteq \mathbb{R}^d$). |
| $\mathbf{x}$ | An input decision vector, $\mathbf{x} \in \mathcal{X}$. |
| $\mathbf{x}^*$ | The global optimum of the target function. |
| $\mathbf{x}_k^*$ | The global optimum of the k-th source task. |
| $K$ | The number of source tasks. |
| $f_k(\cdot)$ | The $k$-th source task function ($k \in \{1, \dots, K\}$). |
| $\mathcal{D}_t$ | The accumulated target dataset at iteration $t$, $\mathcal{D}_t = \{(\mathbf{x}_i, y_i)\}_{i=1}^{n_t}$. |
| $\mathcal{D}_k$ | The static historical dataset for the $k$-th source task, $\mathcal{D}_k = \{(\mathbf{x}_i^k, y_i^k)\}_{i=1}^{n_k}$. |
| $T$ | The total optimization budget. |
| **2. Surrogate Modeling & Ranking** | |
| $F$ | The probabilistic value-based surrogate model for the target task (here a GP). |
| $F_k$ | The rank-based surrogate model for the $k$-th source task (here Deep Ensembles). |
| $V_k$ | The value-based surrogate model for the $k$-th source task (here Deep Ensembles). |
| $\mu_{\hat{k}}(\cdot), \sigma_{\hat{k}}(\cdot), \text{Cov}_{\hat{k}}(\cdot, \cdot)$ | Predictive mean, standard deviation, and covariance functions of $V_k$. |
| $\mu(\cdot), \sigma(\cdot)$ | Predictive mean and standard deviation functions of $F$. |
| $\mu_k(\cdot), \sigma_k(\cdot)$ | Predictive mean and standard deviation functions of the $F_k$. |
| $\mathcal{L}_{pair}$ | Pairwise ranking loss function. |
| $\mathcal{L}_{list}$ | Listwise ranking loss function. |
| $p_{ij}^k$ | Probability that model $V_k$ ranks $\mathbf{x}_i$ higher than $\mathbf{x}_j$. |
| $N_c^k, N_d^k$ | The number of concordant and discordant pairs for the $k$-th source task. |
| $\tau_k$ | Expected Kendall's Rank Correlation Coefficient between source $k$ and the target. |
| $\hat{\tau}_k$ | Empirical estimator of Kendall's $\tau_k$ based on current data. |
| **3. Transfer & Fusion** | |
| $w_k$ | Transfer weight for the $k$-th source task ($w_k = |\tau_k|$). |
| $\mathcal{X}_{norm}$ | Normalization set used to calibrate scales across different tasks. |
| $\mu_{norm}^F, \sigma_{norm}^F$ | Mean and standard deviation statistics of $F$ computed on $\mathcal{X}_{norm}$. |
| $\mu_{norm}^k, \sigma_{norm}^k$ | Mean and standard deviation statistics of $F_k$ computed on $\mathcal{X}_{norm}$. |
| $\tilde{F}$ | The normalized surrogate model for the target task. |
| $\tilde{F}_k$ | The normalized surrogate model for the $k$-th source task. |
| $\tilde{\mu}(\cdot), \tilde{\sigma}(\cdot)$ | Normalized predictive mean and standard deviation functions of $\tilde{F}$. |
| $\tilde{\mu}_k(\cdot), \tilde{\sigma}_k(\cdot)$ | Normalized predictive mean and standard deviation functions of $\tilde{F}_k$. |
| $\tilde{\boldsymbol{\mu}}, \tilde{\boldsymbol{\Sigma}}$ | Normalized predictive mean vector and covariance matrix of $\tilde{F}$ on query set $\mathbf{X}$. |
| $\tilde{\boldsymbol{\mu}}_k, \tilde{\boldsymbol{\Sigma}}_k$ | Normalized predictive mean vector and covariance matrix of $\tilde{F}_k$ on query set $\mathbf{X}$. |
| $F_{fuse}$ | The final fused surrogate model derived via gPoE. |
| $\mu_{fuse}(\cdot), \sigma_{fuse}(\cdot)$ | Predictive mean and standard deviation functions of $F_{fuse}$. |

| | |
|---|---|
| $\boldsymbol{\mu}_{fuse}, \boldsymbol{\Sigma}_{fuse}$ | Predictive mean vector and covariance matrix of $F_{fuse}$ on query set $\mathbf{X}$. |

**4. Acquisition Function (RA-UCB)**

| | |
|---|---|
| $\alpha_t(\cdot)$ | Acquisition function at iteration $t$. |
| $\beta_t$ | Exploration parameter balancing the standard UCB term. |
| $\lambda_t$ | Weight parameter for the Rank Entropy (REN) term. |
| $\text{REN}(\cdot; \cdot)$ | Rank Entropy term, quantifying the ordinal uncertainty relative to anchors. |
| $\mathcal{A}_{src}$ | Set of high-quality anchor points derived from source tasks. |
| $M$ | The number of anchor points selected per source task. |
| $\text{Cov}_{fuse}(\cdot, \cdot)$ | Posterior covariance between a candidate point and an anchor point. |
| $p_{\mathbf{x}, \mathbf{a}}$ | Probability that model $F_{fuse}$ ranks $\mathbf{x}$ higher than $\mathbf{a}$. |
| $H(\cdot)$ | Binary entropy function (approximated as $p(1-p)$). |

**5. Theoretical Analysis**

| | |
|---|---|
| $R_T$ | Cumulative regret of the algorithm over $T$ iterations. |
| $\gamma_T$ | Maximum information gain of the target kernel (theoretical quantity). |
| $L$ | Lipschitz constant of the target function $f$. |
| $\alpha$ | Local growth exponent of $f$ near the global optimum $\mathbf{x}^*$. |
| $\epsilon$ | Estimation error threshold for concentration bounds. |

# B. Derivation of the gPoE-based Fused Posterior

For the sake of completeness and to explicitly demonstrate the adaptation of the Generalized Product of Experts (gPoE) framework (Deisenroth & Ng, 2015) to our specific normalized surrogate setting, this section provides a detailed derivation of the fused predictive mean vector $\boldsymbol{\mu}_{fuse}$ and covariance matrix $\boldsymbol{\Sigma}_{fuse}$ conditioned on the query set $\mathbf{X}$, as presented in Equation (8).

## B.1. Assumptions

Our derivation is grounded in the following two core assumptions:

**Multivariate Gaussian Inputs:**

We assume that the predictive distributions for both the normalized target and source surrogate models over a finite set of query points $\mathbf{X}$ follow Multivariate Gaussian distributions.

As for the normalized target surrogate model, since it is obtained by applying a linear transformation to the fitted Gaussian Process, it naturally satisfies the assumption, as linear operations preserve Gaussianity. As for the normalized source surrogate models, due to the large-scale datasets, we employ Deep Ensembles as a computational approximation to Gaussian Processes and assume they exhibit Gaussian properties (Lakshminarayanan et al., 2017).

Under this assumption, the predictive posteriors of both the normalized target model $\tilde{F}$ and the $k$-th normalized source model $\tilde{F}_k$ on the query set $\mathbf{X}$ are explicitly formulated as Multivariate Gaussian densities:

$$p_{\tilde{F}}(\mathbf{y} \mid \mathbf{X}) = \frac{1}{\sqrt{(2\pi)^N |\tilde{\boldsymbol{\Sigma}}|}} \exp\left(-\frac{1}{2}(\mathbf{y} - \tilde{\boldsymbol{\mu}})^\top \tilde{\boldsymbol{\Sigma}}^{-1}(\mathbf{y} - \tilde{\boldsymbol{\mu}})\right), \tag{15}$$

and

$$p_{\tilde{F}_k}(\mathbf{y} \mid \mathbf{X}) = \frac{1}{\sqrt{(2\pi)^N |\tilde{\boldsymbol{\Sigma}}_k|}} \exp\left(-\frac{1}{2}(\mathbf{y} - \tilde{\boldsymbol{\mu}}_k)^\top \tilde{\boldsymbol{\Sigma}}_k^{-1}(\mathbf{y} - \tilde{\boldsymbol{\mu}}_k)\right), \tag{16}$$

where $\mathbf{y} \in \mathbb{R}^N$ denotes the vector of predicted values, $N$ is the number of query points, and $|\cdot|$ represents the determinant of the covariance matrix.

**gPoE Fusion Rule:** Under the gPoE framework, the probability density function (PDF) of the fused posterior, $p_{fuse}(\mathbf{y}|\mathbf{X})$, is proportional to the product of the target density and the source densities, where each source density is exponentiated by its corresponding trustworthiness weight:

$$p_{fuse}(\mathbf{y}|\mathbf{X}) \propto p_{\tilde{F}}(\mathbf{y}|\mathbf{X}) \prod_{k=1}^{K} \left[ p_{\tilde{F}_k}(\mathbf{y}|\mathbf{X}) \right]^{w_k}, \tag{17}$$

### B.2. Derivation of Fused Posterior Parameters

We derive the closed-form expressions for $\boldsymbol{\mu}_{fuse}$ and $\boldsymbol{\Sigma}_{fuse}$ by completing the square in the logarithmic domain. The probability density function of an $N$-dimensional Multivariate Gaussian $\mathcal{N}(\mathbf{y} \mid \boldsymbol{\mu}, \boldsymbol{\Sigma})$ can be written in its canonical log-form as:

$$\ln p(\mathbf{y}) = \underbrace{-\frac{1}{2}\mathbf{y}^\top \boldsymbol{\Sigma}^{-1}\mathbf{y}}_{\text{Quadratic Term}} + \underbrace{\mathbf{y}^\top (\boldsymbol{\Sigma}^{-1}\boldsymbol{\mu})}_{\text{Linear Term}} + C, \tag{18}$$

where $C$ absorbs terms independent of $\mathbf{y}$. The distribution is uniquely determined by the coefficient of the quadratic term (precision matrix $\boldsymbol{\Sigma}^{-1}$) and the linear term (precision-adjusted mean $\boldsymbol{\Sigma}^{-1}\boldsymbol{\mu}$).

Applying the logarithm to the gPoE fusion rule $p_{fuse}(\mathbf{y}) \propto p_{\tilde{F}}(\mathbf{y}) \prod_k [p_{\tilde{F}_k}(\mathbf{y})]^{w_k}$ yields:

$$\ln p_{fuse}(\mathbf{y}) = \ln p_{\tilde{F}}(\mathbf{y}) + \sum_{k=1}^{K} w_k \ln p_{\tilde{F}_k}(\mathbf{y}) + \text{const.} \tag{19}$$

Substituting Equation (18) into Equation (19) and grouping terms by powers of $\mathbf{y}$:

$$\ln p_{fuse}(\mathbf{y}) = -\frac{1}{2}\mathbf{y}^\top \left( \tilde{\boldsymbol{\Sigma}}^{-1} + \sum_{k=1}^{K} w_k \tilde{\boldsymbol{\Sigma}}_k^{-1} \right) \mathbf{y} + \mathbf{y}^\top \left( \tilde{\boldsymbol{\Sigma}}^{-1}\tilde{\boldsymbol{\mu}} + \sum_{k=1}^{K} w_k \tilde{\boldsymbol{\Sigma}}_k^{-1}\tilde{\boldsymbol{\mu}}_k \right) + \text{const.} \tag{20}$$

By matching these coefficients with the canonical form of the fused posterior $\mathcal{N}(\boldsymbol{\mu}_{fuse}, \boldsymbol{\Sigma}_{fuse})$, we identify the parameters:

**Covariance Matrix.** Matching the quadratic terms gives the fused precision matrix:

$$\boldsymbol{\Sigma}_{fuse}^{-1} = \tilde{\boldsymbol{\Sigma}}^{-1} + \sum_{k=1}^{K} w_k \tilde{\boldsymbol{\Sigma}}_k^{-1} \implies \boldsymbol{\Sigma}_{fuse} = \left( \tilde{\boldsymbol{\Sigma}}^{-1} + \sum_{k=1}^{K} w_k \tilde{\boldsymbol{\Sigma}}_k^{-1} \right)^{-1}. \tag{21}$$

**Mean Vector.** Matching the linear terms gives:

$$\boldsymbol{\Sigma}_{fuse}^{-1}\boldsymbol{\mu}_{fuse} = \tilde{\boldsymbol{\Sigma}}^{-1}\tilde{\boldsymbol{\mu}} + \sum_{k=1}^{K} w_k \tilde{\boldsymbol{\Sigma}}_k^{-1}\tilde{\boldsymbol{\mu}}_k \implies \boldsymbol{\mu}_{fuse} = \boldsymbol{\Sigma}_{fuse} \left( \tilde{\boldsymbol{\Sigma}}^{-1}\tilde{\boldsymbol{\mu}} + \sum_{k=1}^{K} w_k \tilde{\boldsymbol{\Sigma}}_k^{-1}\tilde{\boldsymbol{\mu}}_k \right). \tag{22}$$

When considering a single query point or assuming independence between query points, the matrix operations degenerate into scalar operations. In this setting, the precision matrix becomes the reciprocal of the variance. Consequently, the fusion rule simplifies to the standard inverse-variance weighting mechanism:

$$\frac{1}{\sigma_{fuse}^2} = \frac{1}{\tilde{\sigma}^2} + \sum_{k=1}^{K} \frac{w_k}{\tilde{\sigma}_k^2}, \quad \text{and} \quad \mu_{fuse} = \sigma_{fuse}^2 \left( \frac{\tilde{\mu}}{\tilde{\sigma}^2} + \sum_{k=1}^{K} w_k \frac{\tilde{\mu}_k}{\tilde{\sigma}_k^2} \right). \tag{23}$$

This scalar form demonstrates that predictions with lower variance (higher precision) and higher trustworthiness weights $w_k$ exert the most significant influence on the final fused estimate.

## C. Proofs of Concentration Bounds for Rank Correlation

In this section, we provide the theoretical justification for utilizing the empirical Kendall's rank correlation coefficient, Kendall's $\tau$, to quantify task similarity. We start our analysis by characterizing Kendall's $\tau$ within the U-statistic framework and employ the **Hoeffding decomposition** to analyze its structure. Based on this decomposition, we can derive two concentration bounds: a **General Bound** via Hoeffding's inequality and a tighter **Adaptive Bound** via Bernstein's inequality, which demonstrates accelerated convergence when tasks are highly correlated ($\tau \to 1$). Finally, we briefly discuss the extension to sequential settings, where observations are inherently dependent as the selection of query points is conditioned on the accumulated history.

### C.1. Hoeffding Decomposition of U-Statistics

For any arbitrary source model $k \in \{1, \ldots, K\}$, let $\mathcal{D}_{\text{init}} = \{(\mathbf{x}_i, y_i)\}_{i=1}^{n_{\text{init}}}$ denote the set of $n_{\text{init}}$ independent and identically distributed (i.i.d.) observations collected during the initialization phase. To rigorously analyze the rank correlation while accounting for predictive uncertainty, we define the model-specific analysis variable as $\mathbf{z}_{i,k} = (y_i, V_k(\mathbf{x}_i))$, where $V_k(\mathbf{x}_i) \sim \mathcal{N}(\mu_k(\mathbf{x}_i), \sigma_k^2(\mathbf{x}_i))$ represents the full predictive distribution provided by the $k$-th source model.

We define the population Kendall's $\tau_k$ and its empirical U-statistic estimator $\hat{\tau}_k$ for model $k$ with sample size $n = n_{\text{init}}$ as:

$$\tau_k = \mathbb{E}_{\mathbf{z}, \mathbf{z}' \sim \mathcal{P}}[h_k(\mathbf{z}, \mathbf{z}')], \quad \hat{\tau}_k = \frac{2}{n(n-1)} \sum_{1 \leq i < j \leq n} h_k(\mathbf{z}_{i,k}, \mathbf{z}_{j,k}), \tag{24}$$

where $\mathbf{z}$ and $\mathbf{z}'$ are independent copies drawn from the joint distribution $\mathcal{P}$. The kernel function $h_k$ is constructed as:

$$h_k(\mathbf{z}_{i,k}, \mathbf{z}_{j,k}) = \text{sgn}(y_i - y_j) \cdot (2p_{ij}^k - 1), \tag{25}$$

where $p_{ij}^k = \mathbb{P}(V_k(\mathbf{x}_i) > V_k(\mathbf{x}_j))$ is the ranking probability. Here, $h_k$ quantifies the probabilistic concordance between the predicted ranking of model $k$ and the ground-truth order.

Since the kernel $h_k$ is symmetric and bounded within $[-1, 1]$, the estimator $\hat{\tau}_k$ constitutes a U-statistic of order two. To analyze its asymptotic structure, we apply the **Hoeffding Decomposition** (Hoeffding, 1992) to linearize $\hat{\tau}_k$ as follows:

$$\hat{\tau}_k - \tau_k = \frac{2}{n} \sum_{i=1}^{n} \hat{h}_{1,k}(\mathbf{z}_{i,k}) + R_{n,k}. \tag{26}$$

The linear interaction is captured by the first-order Hájek projection $\hat{h}_{1,k}$, which represents the conditional expectation of the kernel:

$$\hat{h}_{1,k}(\mathbf{z}_{i,k}) = \mathbb{E}[h_k(\mathbf{z}_{i,k}, \mathbf{z}_{j,k}) \mid \mathbf{z}_{i,k}] - \tau_k. \tag{27}$$

By construction, this term is centered, i.e., $\mathbb{E}[\hat{h}_{1,k}(\mathbf{z}_{i,k})] = 0$. The remaining term, $R_{n,k}$, denotes the second-order degenerate residual:

$$R_{n,k} = \binom{n}{2}^{-1} \sum_{1 \leq i < j \leq n} \psi_k(\mathbf{z}_{i,k}, \mathbf{z}_{j,k}), \tag{28}$$

where $\psi_k(\mathbf{z}_{i,k}, \mathbf{z}_{j,k}) = h_k(\mathbf{z}_{i,k}, \mathbf{z}_{j,k}) - \hat{h}_{1,k}(\mathbf{z}_{i,k}) - \hat{h}_{1,k}(\mathbf{z}_{j,k}) - \tau_k$ is the canonical degenerate kernel. The decomposition demonstrates that the variance of the linear projection scales as $\mathcal{O}(n^{-1})$, whereas the residual $R_{n,k}$ decays at a faster rate of $\mathcal{O}(n^{-2})$. In the proof of the adaptive bound, we focus on the dominant linear sum $\sum \hat{h}_{1,k}(\mathbf{z}_{i,k})$.

### C.2. Proof of General Bound (Hoeffding)

To derive the General Bound for the empirical estimator $\hat{\tau}_k$, we directly apply the Hoeffding inequality established for U-statistics (Hoeffding, 1963). For a second-order U-statistic $\hat{\tau}_k$ with a kernel $h(\cdot, \cdot)$ bounded in $[a, b]$, the generalized Hoeffding's inequality guarantees:

$$\mathbb{P}(|\hat{\tau}_k - \tau_k| \geq \epsilon) \leq 2 \exp\left(-\frac{2\lfloor n/2 \rfloor \epsilon^2}{(b-a)^2}\right). \tag{29}$$

In our application, the kernel is defined in Equation (25), which is bounded in $[-1, 1]$. Substituting these values into Equation (29) yields:

$$\mathbb{P}(|\hat{\tau}_k - \tau_k| \geq \epsilon) \leq 2 \exp\left(-\frac{2\lfloor n/2 \rfloor \epsilon^2}{4}\right) = 2 \exp\left(-\frac{\lfloor n/2 \rfloor \epsilon^2}{2}\right). \tag{30}$$

By applying the asymptotic approximation $\lfloor n/2 \rfloor \approx n/2$, the bound simplifies to the form presented in Theorem 5.1:

$$\mathbb{P}(|\hat{\tau}_k - \tau_k| \geq \epsilon) \leq 2 \exp\left(-\frac{n\epsilon^2}{4}\right). \tag{31}$$

## C.3. Proof of Adaptive Bound (Bernstein)

To derive the variance-adaptive bound for the $k$-th source model, we only focus on the linear component $\hat{h}_{1,k}(\mathbf{z}_{i,k})$ of the Hoeffding decomposition in Equation (26). Let $\sigma_{h,k}^2 = \mathrm{Var}(\hat{h}_{1,k}(\mathbf{z}_{i,k}))$ denote the variance of the first-order projection for model $k$. Because the kernel function $h_k$ is bounded in $[-1, 1]$, its second moment satisfies $\mathbb{E}[h_k^2] \leq 1$. This implies that the total variance of the kernel is bounded: $\mathrm{Var}(h_k) = \mathbb{E}[h_k^2(\mathbf{z}_{i,k}, \mathbf{z}_{j,k})] - \mathbb{E}[h_k(\mathbf{z}_{i,k}, \mathbf{z}_{j,k})]^2 = \mathbb{E}[h_k^2(\mathbf{z}_{i,k}, \mathbf{z}_{j,k})] - \tau_k^2 \leq 1 - \tau_k^2$. By the Law of Total Variance, the variance of the conditional expectation (the projection) cannot exceed the total variance of the original kernel:

$$\begin{aligned}
\sigma_{h,k}^2 &= \mathrm{Var}\left(\mathbb{E}[h_k(\mathbf{z}_{i,k}, \mathbf{z}_{j,k}) \mid \mathbf{z}_{i,k}]\right) \\
&= \mathrm{Var}(h_k(\mathbf{z}_{i,k}, \mathbf{z}_{j,k})) - \mathbb{E}\left[\mathrm{Var}(h_k \mid \mathbf{z}_{i,k})\right] \\
&\leq \mathrm{Var}(h_k(\mathbf{z}_{i,k}, \mathbf{z}_{j,k})) \\
&= \mathbb{E}[h_k^2(\mathbf{z}_{i,k}, \mathbf{z}_{j,k})] - (\mathbb{E}[h_k(\mathbf{z}_{i,k}, \mathbf{z}_{j,k})])^2 \\
&\leq 1 - \tau_k^2.
\end{aligned} \tag{32}$$

This property means that as the $k$-th source task becomes highly similar to the target task ($|\tau_k| \to 1$), the variance of the estimator vanishes.

To establish the concentration bound, we use Bernstein's inequality.

**Lemma C.1** (Bernstein's Inequality). *Let $X_1, \ldots, X_n$ be independent zero-mean random variables. Suppose that $|X_i| \leq M$ and $\frac{1}{n} \sum_{i=1}^n \mathbb{E}[X_i^2] \leq \sigma^2$. Then, for any $t > 0$, the following concentration inequality holds:*

$$\mathbb{P}\left(\left|\frac{1}{n} \sum_{i=1}^n X_i\right| \geq t\right) \leq 2 \exp\left(-\frac{nt^2}{2\sigma^2 + \frac{2}{3}Mt}\right). \tag{33}$$

Set $X_i = \hat{h}_{1,k}(\mathbf{z}_{i,k})$ in Equation (27) and $t = \epsilon/2$, and notice that the centered variables $\hat{h}_{1,k}$ are bounded by $M = 2$ and $\frac{1}{n} \sum_{i=1}^n \mathbb{E}[\hat{h}_{1,k}(\mathbf{z}_{i,k})^2] = \frac{1}{n} \sum_{i=1}^n \mathrm{Var}(\hat{h}_{1,k}(\mathbf{z}_{i,k})) = \sigma_{h,k}^2 \leq 1 - \tau_k^2$. We can obtain:

$$\mathbb{P}\left(\left|\frac{2}{n} \sum_{i=1}^n \hat{h}_{1,k}(\mathbf{z}_{i,k})\right| \geq \epsilon\right) = \mathbb{P}\left(\left|\frac{1}{n} \sum_{i=1}^n \hat{h}_{1,k}(\mathbf{z}_{i,k})\right| \geq \frac{\epsilon}{2}\right) \leq 2 \exp\left(-\frac{n(\epsilon/2)^2}{2\sigma_{h,k}^2 + \frac{2}{3}M(\epsilon/2)}\right). \tag{34}$$

Substituting the variance bound $\sigma_{h,k}^2 \leq 1 - \tau_k^2$, we obtain the Adaptive Bound:

$$\mathbb{P}(|\hat{\tau}_k - \tau_k| \geq \epsilon) \leq 2 \exp\left(-\frac{n\epsilon^2}{8(1 - \tau_k^2) + C\epsilon}\right). \tag{35}$$

where the constant $C = 8/3$.

## C.4. Extension to Sequential Settings

Our analysis above is based on the i.i.d. assumption. However, in the optimization phase, new samples depend on historical data and fail to satisfy the requirement. We note that our results can be naturally extended to this sequential setting via Martingale theory. We can recover a similar exponential convergence rate, the **Azuma-Hoeffding inequality** for martingales with bounded differences. Crucially, in transfer learning, source tasks with higher correlation are of significantly greater value. Our theoretical analysis highlights the **Bernstein bound** in the high-similarity regime ($\tau_k \to 1$), where the variance reduction effect accelerates convergence, which is essential for efficient transfer.

# D. Proofs of Optimization Consistency

In this section, we address the fundamental question in Theorem 5.2: How does Kendall's Rank Correlation Coefficient $\tau_k$ characterize the spatial proximity between the source optimal solution $\mathbf{x}_k^*$ and the target optimal solution $\mathbf{x}^*$?

## D.1. Assumptions

To bridge the gap between ranking consistency and spatial proximity, we introduce geometric assumptions. Unlike the simplified case, we acknowledge that the source and target tasks may possess distinct landscapes. Let $\mathcal{X} \subset \mathbb{R}^d$ be a compact domain.

**Assumption D.1** (Lipschitz Continuity). We assume both tasks are Lipschitz continuous with different constants. The target function $f$ is $L$-Lipschitz continuous, and the source function $f_k$ is $L_k$-Lipschitz continuous. For any $\mathbf{x}, \mathbf{y} \in \mathcal{X}$:

$$|f(\mathbf{x}) - f(\mathbf{y})| \leq L\|\mathbf{x} - \mathbf{y}\|, \quad |f_k(\mathbf{x}) - f_k(\mathbf{y})| \leq L_k\|\mathbf{x} - \mathbf{y}\|. \tag{36}$$

**Assumption D.2** (Unique Global Optima and Local Growth). We assume both tasks have unique global optima ($\mathbf{x}^*$ and $\mathbf{x}_k^*$) on the compact domain $\mathcal{X}$. The target function $f$ satisfies a local growth condition within a neighborhood $B(\mathbf{x}^*, \delta)$ and maintains a separation constant outside. specifically, with parameters $\mu, \delta, \epsilon > 0$ and $\alpha \geq 1$:

$$f(\mathbf{x}^*) - f(\mathbf{x}) \geq \begin{cases} \mu\|\mathbf{x} - \mathbf{x}^*\|^\alpha, & \text{if } \|\mathbf{x} - \mathbf{x}^*\| \leq \delta \quad \text{(Local Growth)}, \\ \epsilon, & \text{if } \|\mathbf{x} - \mathbf{x}^*\| > \delta \quad \text{(Global Separation)}. \end{cases} \tag{37}$$

The source function $f_k$ satisfies analogous conditions with parameters $\mu_k, \delta_k, \epsilon_k > 0$ and $\alpha_k \geq 1$.

## D.2. Proof of Theorem 5.2

We restate the theorem from Section 5.2 for clarity.

**Theorem 5.2** (Optimization Consistency). *Let $\tau_k$ be the Kendall's Rank Correlation Coefficient between the $k$-th source task $f_k$ and the target task $f$. Under Assumptions D.1 and D.2, if $\tau_k \to 1$, then the distance between the source optimizer $\mathbf{x}_k^*$ and the target optimizer $\mathbf{x}^*$ converges to zero:*

$$\lim_{\tau_k \to 1} \|\mathbf{x}_k^* - \mathbf{x}^*\| = 0. \tag{38}$$

*Furthermore, the positional error is explicitly bounded by:*

$$\|\mathbf{x}_k^* - \mathbf{x}^*\| \leq C \cdot (1 - \tau_k)^{\frac{1}{2d\bar{\alpha}}}, \tag{39}$$

*where $\bar{\alpha} = \max(\alpha, \alpha_k)$, and $C > 0$ is a constant depending on the geometric parameters $(L, \mu, \alpha, L_k, \mu_k, \alpha_k)$ and the volume of the search space $\mathcal{X}$.*

*Proof.* Firstly, we employ a proof by contradiction to establish that $\lim_{\tau_k \to 1} \|\mathbf{x}_k^* - \mathbf{x}^*\| = 0$. Suppose that as the ranking correlation approaches unity ($\tau_k \to 1$), the source optimal solution $\mathbf{x}_k^*$ remains separated from the target optimal solution $\mathbf{x}^*$ and exhibits a persistent positional error $\|\mathbf{x}_k^* - \mathbf{x}^*\| = r > 0$.

**Constraints from the Target Task:** According to Assumption D.2, since $r > 0$ and the optimum is unique, there exists a strictly positive value gap $\Delta = f(\mathbf{x}^*) - f(\mathbf{x}_k^*) > 0$. We construct two hyperspheres: Region $A = B(\mathbf{x}^*, \rho)$ centered at the target optimum, and Region $B = B(\mathbf{x}_k^*, \rho)$ centered at the source optimum. To guarantee spatial disjointness, we first require $\rho < r/2$. Next, to ensure that every point in $A$ strictly outperforms any point in $B$ on the target function (i.e., $f(\mathbf{u}) > f(\mathbf{v})$ for all $\mathbf{u} \in A, \mathbf{v} \in B$), we utilize the Lipschitz continuity of $f$:

$$\min_{\mathbf{u} \in A} f(\mathbf{u}) - \max_{\mathbf{v} \in B} f(\mathbf{v}) \geq (f(\mathbf{x}^*) - L\rho) - (f(\mathbf{x}_k^*) + L\rho) = \Delta - 2L\rho. \tag{40}$$

Requiring this difference to be positive imposes the constraint: $\rho < \frac{\Delta}{2L}$.

**Constraints from the Source Task:** Simultaneously, to form discordant pairs, the ranking must be **consistent with the source task** (i.e., $f_k(\mathbf{v}) > f_k(\mathbf{u})$), effectively reversing the preference order. Since $\mathbf{x}_k^*$ is the unique global optimum for the

source task, there exists an analogous value gap $\Delta_k = f_k(\mathbf{x}_k^*) - f_k(\mathbf{x}^*) > 0$. By applying the same Lipschitz logic ($L_k$) symmetrically, strict separation on the source task requires:

$$(f_k(\mathbf{x}_k^*) - L_k\rho) - (f_k(\mathbf{x}^*) + L_k\rho) = \Delta_k - 2L_k\rho > 0. \tag{41}$$

This imposes the corresponding constraint: $\rho < \frac{\Delta_k}{2L_k}$.

Combining the constraints from both tasks and the spatial requirement, we require:

$$\rho < \min\left(\frac{r}{2}, \frac{\Delta}{2L}, \frac{\Delta_k}{2L_k}\right). \tag{42}$$

For any fixed $r > 0$, since $\Delta, L, \mu_k, L_k$ are all positive constants, the right-hand side is a strictly positive value. Therefore, there always exists a sufficiently small radius $\rho > 0$ satisfying this condition. With such a radius, regions $A$ and $B$ are disjoint, and the ranking is strictly inverted ($f(\mathbf{u}) > f(\mathbf{v})$ but $f_k(\mathbf{v}) > f_k(\mathbf{u})$). Consequently, every pair $(\mathbf{u}, \mathbf{v}) \in A \times B$ constitutes a discordant pair. Based on Equation (3), we obtain:

$$\frac{1 - \tau_k}{2} \geq \frac{Vol(A) \cdot Vol(B)}{Vol(\mathcal{X})^2} \propto \rho^{2d}. \tag{43}$$

Since $\rho$ is a fixed positive number, the lower bound in Equation (43) is a strictly positive constant. This stands in contradiction to the premise that $\tau_k \to 1$, which requires the term $\frac{1-\tau_k}{2}$ to approach zero. Consequently, the initial hypothesis must be false, and we conclude that $\lim_{\tau_k \to 1} \|\mathbf{x}_k^* - \mathbf{x}^*\| = 0$. $\qquad\square$

Next, we prove that as $\tau_k \to 1$, the following bound holds:

$$\|\mathbf{x}_k^* - \mathbf{x}^*\| \leq C \cdot (1 - \tau_k)^{\frac{1}{2d\bar{\alpha}}}. \tag{44}$$

*Proof.* Having established convergence, we assume that when $\tau_k \to 1$, the source optimizer $\mathbf{x}_k^*$ lies within the local neighborhood $\mathcal{N}(\mathbf{x}^*)$ where the Local $\alpha$-Order Growth condition holds. Let $r = \|\mathbf{x}^* - \mathbf{x}_k^*\|$ denote the positional error.

To derive the explicit bound, we utilize the unified radius construction:

$$\rho = \frac{1}{M}r^{\bar{\alpha}}, \quad \text{where } \bar{\alpha} = \max(\alpha, \alpha_k). \tag{45}$$

We assume $M$ is sufficiently large to ensure spatial disjointness ($M > 2r^{\bar{\alpha}-1}$, implying $\rho < r/2$). Next, we derive the constraints on $M$ via functional separation.

**Separation on Target Task $f$:** We require strict separation $f(\mathbf{u}) > f(\mathbf{v})$ for any pair $(\mathbf{u}, \mathbf{v}) \in A \times B$.

- **Lower Bound on $A$:** For any $\mathbf{u} \in B(\mathbf{x}^*, \rho)$, by Lipschitz continuity:

$$f(\mathbf{u}) \geq f(\mathbf{x}^*) - L\rho.$$

- **Upper Bound on $B$:** For any $\mathbf{v} \in B(\mathbf{x}_k^*, \rho)$, by Lipschitz continuity and the growth condition at $\mathbf{x}_k^*$:

$$f(\mathbf{v}) \leq f(\mathbf{x}_k^*) + L\rho \leq (f(\mathbf{x}^*) - \mu r^\alpha) + L\rho.$$

Combining these, the value gap is bounded by:

$$f(\mathbf{u}) - f(\mathbf{v}) \geq (f(\mathbf{x}^*) - L\rho) - (f(\mathbf{x}^*) - \mu r^\alpha + L\rho) = \mu r^\alpha - 2L\rho. \tag{46}$$

To ensure positivity, we substitute $\rho = \frac{1}{M}r^{\bar{\alpha}}$:

$$\mu r^\alpha > \frac{2L}{M}r^{\bar{\alpha}} \implies M > \frac{2L}{\mu}r^{\bar{\alpha}-\alpha}. \tag{47}$$

**Separation on Source Task $f_k$:** Symmetrically, we require the inverse ranking $f_k(\mathbf{v}) > f_k(\mathbf{u})$.

- **Lower Bound on $B$:** For any $\mathbf{v} \in B(\mathbf{x}_k^*, \rho)$, by Lipschitz continuity:

$$f_k(\mathbf{v}) \geq f_k(\mathbf{x}_k^*) - L_k\rho.$$

- **Upper Bound on $A$:** For any $\mathbf{u} \in B(\mathbf{x}^*, \rho)$, since $\mathbf{u}$ is near $\mathbf{x}^*$ (distance $r$), we apply the growth condition centered at $\mathbf{x}_k^*$. Approximating the distance $\|\mathbf{x}^* - \mathbf{x}_k^*\| = r$:

$$f_k(\mathbf{u}) \leq f_k(\mathbf{x}^*) + L_k\rho \leq (f_k(\mathbf{x}_k^*) - \mu_k r^{\alpha_k}) + L_k\rho.$$

The value gap is bounded by:

$$f_k(\mathbf{v}) - f_k(\mathbf{u}) \geq (f_k(\mathbf{x}_k^*) - L_k\rho) - (f_k(\mathbf{x}_k^*) - \mu_k r^{\alpha_k} + L_k\rho) = \mu_k r^{\alpha_k} - 2L_k\rho. \tag{48}$$

To ensure positivity, we substitute $\rho = \frac{1}{M}r^{\bar{\alpha}}$:

$$\mu_k r^{\alpha_k} > \frac{2L_k}{M}r^{\bar{\alpha}} \implies M > \frac{2L_k}{\mu_k}r^{\bar{\alpha}-\alpha_k}. \tag{49}$$

**Deriving the Explicit Bound:** To satisfy the requirements imposed by spatial disjointness, target functional separation, and source functional separation simultaneously, we choose $M$ such that:

$$M > \max\left(2r^{\bar{\alpha}-1}, \ \frac{2L}{\mu}r^{\bar{\alpha}-\alpha}, \ \frac{2L_k}{\mu_k}r^{\bar{\alpha}-\alpha_k}\right). \tag{50}$$

With this sufficiently large $M$, strict separation holds, and every pair $(\mathbf{u}, \mathbf{v}) \in A \times B$ is discordant. Applying the volume inequality:

$$\frac{1-\tau_k}{2} \geq \frac{(V_d\rho^d)^2}{\text{Vol}(\mathcal{X})^2} = \left(\frac{V_d}{\text{Vol}(\mathcal{X})}\right)^2 \left(\frac{1}{M}r^{\bar{\alpha}}\right)^{2d}. \tag{51}$$

Rearranging to solve for $r$, we obtain the bound:

$$\|\mathbf{x}^* - \mathbf{x}_k^*\| \leq \underbrace{\left[M\left(\frac{\text{Vol}(\mathcal{X})}{\sqrt{2}V_d}\right)^{\frac{1}{d}}\right]^{\frac{1}{\bar{\alpha}}}}_{C} \cdot (1-\tau_k)^{\frac{1}{2d\bar{\alpha}}}. \tag{52}$$

where $V_d = \frac{\pi^{d/2}}{\Gamma(d/2+1)}$ denotes the volume of the unit hypersphere in $\mathbb{R}^d$, and $\Gamma(\cdot)$ is the Gamma function. $\qquad\square$

*Remark* D.3 (Geometric Interpretation of Convergence Rate). The derived bound in Equation (52) also reveals how the problem geometry and dimensionality affect the difficulty of optimization consistency via ranking.

- **Impact of Dimensionality ($d$):** As the dimension $d$ increases, the convergence bound expands and the error tolerance decreases. Achieving the same spatial precision imposes a stricter requirement on ranking consistency.

- **Impact of Function Volatility ($L$,$L_k$):** As the Lipschitz constants $L$ and $L_k$ increase, the function fluctuations become more drastic and impose a stricter requirement on ranking consistency.

- **Impact of Peak Sharpness ($\mu$,$\mu_k$):** As the growth parameter $\mu$ and $mu_k$ increase, the global optimum becomes more distinct and distinguishable, thereby relaxing the requirement on ranking consistency.

*Remark* D.4 (Interpretation of Reliability). The derived bound in Equation (52) establishes a theoretical link between the Kenall correlation $\tau_k$ and optimization precision. It suggests that a high $\tau_k$ is a necessary condition for a tight error bound on the source optimum. In contrast, a lower $\tau_k$ fails to yield a tight bound, thereby lacking theoretical guarantees for spatial proximity. This validates the reliability of our framework: we assign higher confidence to source tasks with higher $\tau_k$, while mitigating the risk associated with lower $\tau_k$.

## E. Regret Bounds and Convergence Analysis

In this section, we provide the theoretical analysis of the RA-UCB and prove its no-regret property despite the introduction of the REN term.

Let $x^* = \arg\max_{x \in \mathcal{D}} f(x)$ denote the global optimum. The instantaneous regret at step $t$ is defined as $r_t = f(x^*) - f(x_t)$. According to standard GP, with probability at least $1 - \delta$, the true function value $f(x)$ falls in the confidence interval:

$$|f(x) - \mu_{t-1}(x)| \leq \beta_t^{1/2}\sigma_{t-1}(x). \tag{53}$$

We define the upper confidence bound as $U_t(x) = \mu_{t-1}(x) + \beta_t^{1/2}\sigma_{t-1}(x)$. The following inequalities hold:

$$f(x) \leq U_t(x) \quad \text{and} \quad f(x) \geq U_t(x) - 2\beta_t^{1/2}\sigma_{t-1}(x). \tag{54}$$

The acquisition function of RA-UCB is defined as $\alpha_t(x) = U_t(x) + \lambda_t\text{REN}_t(x)$, where $0 \leq \text{REN}_t(x) \leq \frac{1}{4}$ (in Equation (11)). Since the algorithm selects $x_t = \arg\max_x \alpha_t(x)$ at step $t$, it must satisfy $\alpha_t(x_t) \geq \alpha_t(x^*)$. Expanding this inequality yields:

$$U_t(x_t) + \lambda_t\text{REN}_t(x_t) \geq U_t(x^*) + \lambda_t\text{REN}_t(x^*). \tag{55}$$

Given that $U_t(x^*) \geq f(x^*)$ and $\text{REN}_t(x^*) \geq 0$, we can derive an upper bound for the optimal value $f(x^*)$:

$$f(x^*) \leq U_t(x^*) \leq U_t(x_t) + \lambda_t\text{REN}_t(x_t) - \lambda_t\text{REN}_t(x^*) \leq U_t(x_t) + \lambda_t\text{REN}_t(x_t). \tag{56}$$

Substituting this upper bound into the definition of instantaneous regret $r_t$, and utilizing the lower bound of $f(x_t)$ from Equation (54), we obtain:

$$\begin{aligned}
r_t &= f(x^*) - f(x_t) \\
&\leq [U_t(x_t) + \lambda_t\text{REN}_t(x_t)] - [U_t(x_t) - 2\beta_t^{1/2}\sigma_{t-1}(x_t)] \\
&= 2\beta_t^{1/2}\sigma_{t-1}(x_t) + \lambda_t\text{REN}_t(x_t).
\end{aligned} \tag{57}$$

This decomposition reveals that the regret consists of two components: the uncertainty term from the standard GP-UCB and the exploration term introduced by the ranking mechanism. Summing over $T$ iterations gives the cumulative regret $R_T$:

$$R_T = \sum_{t=1}^{T} r_t \leq \underbrace{\sum_{t=1}^{T} 2\beta_t^{1/2}\sigma_{t-1}(x_t)}_{\text{Term I}} + \underbrace{\sum_{t=1}^{T} \lambda_t\text{REN}_t(x_t)}_{\text{Term II}}. \tag{58}$$

**Term I Analysis:** This term corresponds to the cumulative reduction in uncertainty. Strictly speaking, our fused model involves **adaptive** weights $w_k$ during optimization. However, we operate under the Stabilising Weights Assumption, which posits that the weights $w_k$ converge to a stable state after a finite period $T_{\text{stab}}$ due to the fixed nature of source tasks. Then the cumulative regret for Term I can be decomposed into two parts: a finite constant term representing the regret accumulated during the adaptive phase ($t \leq T_{\text{stab}}$), and a dominant term governed by the stabilised composite kernel ($t > T_{\text{stab}}$). For the latter phase, the analysis framework of (Srinivas et al., 2010) applies directly. This term is bounded by the maximum information gain $\gamma_T$, and its growth order is dominated by $O(\sqrt{T\beta_T\gamma_T}) \approx O(\sqrt{T})$, which is sub-linear.

**Term II Analysis:** This term represents the additional regret incurred by the rank entropy. To ensure convergence, we must design $\lambda_t$ such that $\sum \lambda_t$ is also sub-linear. We adopt a decay strategy $\lambda_t = Ct^{-1/2}$ (where $C > 0$). Using the integral approximation to estimate the series sum:

$$\sum_{t=1}^{T} \lambda_t = \sum_{t=1}^{T} \frac{C}{\sqrt{t}} \leq C\left(1 + \int_1^T \tau^{-1/2}d\tau\right) = C(1 + 2\sqrt{T} - 2) \leq 2C\sqrt{T}. \tag{59}$$

Consequently, the upper bound for Term II is $\frac{C}{2}\sqrt{T} = O(\sqrt{T})$. Combining both terms, the total cumulative regret satisfies $R_T \leq O(\sqrt{T})$. Finally, the average regret converges to zero:

$$\lim_{T \to \infty} \frac{R_T}{T} = 0. \tag{60}$$

Thus, we prove that the RA-UCB algorithm is No-Regret and converges to the global optimum.

## F. Experimental Details and Additional Analysis

In this section, we provide a detailed description of the synthetic benchmarks, the comprehensive experimental settings, and the additional experiments referenced in Section 6.

### F.1. Benchmark Functions and Source Task Generation

We employed five standard functions for evaluation. The definitions and search spaces are detailed below. All functions are treated as maximization problems in our experiments (by negating the values for standard minimization functions). All these functions exhibit multimodal landscapes with multiple local optima.

**1. Ackley Function ($d = 2, 50$)**

$$f(\mathbf{x}) = -20 \exp\left(-0.2\sqrt{\frac{1}{d}\sum_{i=1}^{d} x_i^2}\right) - \exp\left(\frac{1}{d}\sum_{i=1}^{d}\cos(2\pi x_i)\right) + 20 + e \tag{61}$$

The search space is defined as $\mathbf{x} \in [-32.768, 32.768]^d$ for $d = 2$, whereas for the high-dimensional case ($d = 50$), it is restricted to $\mathbf{x} \in [-5, 5]^d$. The global optimum is located at $\mathbf{x}^* = \mathbf{0}$ with $f(\mathbf{x}^*) = 0$.

**2. Branin Function ($d = 2$)**

The Branin function has three global optima. We negate the standard form:

$$f(\mathbf{x}) = -\left(a(x_2 - bx_1^2 + cx_1 - r)^2 + s(1 - t)\cos(x_1) + s\right) \tag{62}$$

where $a = 1, b = 5.1/(4\pi^2), c = 5/\pi, r = 6, s = 10, t = 1/(8\pi)$. The domain is $x_1 \in [-5, 10], x_2 \in [0, 15]$. The three global maxima are located at $\mathbf{x}^* \in \{(-\pi, 12.275), (\pi, 2.275), (9.42478, 2.475)\}$, with a function value of $f(\mathbf{x}^*) = -0.39$.

**3. Schwefel Function ($d = 6$)**

$$f(\mathbf{x}) = -\left(418.9829d - \sum_{i=1}^{d} x_i \sin(\sqrt{|x_i|})\right) \tag{63}$$

The search space is $\mathbf{x} \in [-500, 500]^d$. The global optimum is at $\mathbf{x}^* = (420.9687, \dots)$ with value 0.

**4. Hartmann 3-Dimensional Function ($d = 3$)**

$$f(\mathbf{x}) = \sum_{i=1}^{4} \alpha_i \exp\left(-\sum_{j=1}^{3} A_{ij}(x_j - P_{ij})^2\right) \tag{64}$$

The search space is the unit hypercube $\mathbf{x} \in (0, 1)^3$. The coefficients are defined as follows:

$$\boldsymbol{\alpha} = (1.0, 1.2, 3.0, 3.2)^T$$

$$\mathbf{A} = \begin{pmatrix} 3.0 & 10 & 30 \\ 0.1 & 10 & 35 \\ 3.0 & 10 & 30 \\ 0.1 & 10 & 35 \end{pmatrix} \tag{65}$$

$$\mathbf{P} = 10^{-4}\begin{pmatrix} 3689 & 1170 & 2673 \\ 4699 & 4387 & 7470 \\ 1091 & 8732 & 5547 \\ 381 & 5743 & 8828 \end{pmatrix}$$

The global maximum is located at $\mathbf{x}^* \approx (0.114614, 0.555649, 0.852547)$ with a value of $f(\mathbf{x}^*) \approx 3.86278$.

**5. Hartmann 6-Dimensional Function ($d = 6$)**

$$f(\mathbf{x}) = \sum_{i=1}^{4} \alpha_i \exp\left(-\sum_{j=1}^{6} A_{ij}(x_j - P_{ij})^2\right) \tag{66}$$

The domain is the unit hypercube $\mathbf{x} \in (0,1)^6$. The coefficients are given by:

$$\boldsymbol{\alpha} = (1.0, 1.2, 3.0, 3.2)^T$$

$$\mathbf{A} = \begin{pmatrix} 10 & 3 & 17 & 3.50 & 1.7 & 8 \\ 0.05 & 10 & 17 & 0.1 & 8 & 14 \\ 3 & 3.5 & 1.7 & 10 & 17 & 8 \\ 17 & 8 & 0.05 & 10 & 0.1 & 14 \end{pmatrix} \tag{67}$$

$$\mathbf{P} = 10^{-4} \begin{pmatrix} 1312 & 1696 & 5569 & 124 & 8283 & 5886 \\ 2329 & 4135 & 8307 & 3736 & 1004 & 9991 \\ 2348 & 1451 & 3522 & 2883 & 3047 & 6650 \\ 4047 & 8828 & 8732 & 5743 & 1091 & 381 \end{pmatrix}$$

The global maximum is located at $\mathbf{x}^* \approx (0.20169, 0.150011, 0.476874, 0.275332, 0.311652, 0.6573)$ with a value of $f(\mathbf{x}^*) \approx 3.32237$.

**Source Task Generation Strategy**

To simulate realistic transfer learning scenarios where source tasks share structural similarities with the target task but differ in optimal locations, we employ a generation strategy that combines *Input Space Shifting* and *Output Space Scaling*.

Let $f_{\text{target}}(\mathbf{x})$ denote the objective function of the target task defined on the bounded domain $[\mathbf{l}, \mathbf{u}]^d$, where $\mathbf{l}$ and $\mathbf{u}$ represent the lower and upper bounds. A source task $f_k(\mathbf{x})$ is constructed via the following transformation:

$$f_k(\mathbf{x}) = \alpha_k \cdot f_{\text{target}}\left(\mathcal{C}\left(\mathbf{x} + \delta_k \cdot (\mathbf{u} - \mathbf{l})\right)\right) \tag{68}$$

where:

- $\delta_k$ represents a translation of the landscape relative to the domain width.

- $\mathcal{C}(\cdot)$ is a Clamping Function that ensures that the shifted input remains within $[\mathbf{l}, \mathbf{u}]$.

- $\alpha_k$ is the Scale Factor. Notably, we use both positive and negative scalars (to vary amplitude) and negative scalars (to invert the landscape) to test the algorithm's ability to handle negative correlations and inverse structural relationships.

The specific generation parameters used in our experiments for the high-dimensional function (Ackley50, $d = 50$) and the low-dimensional functions are detailed in Table 3. The target task always corresponds to the standard configuration with $\alpha = 1.0$ and $\delta = 0.0$.

*Table 3.* Configuration of Source Tasks for Synthetic Benchmarks.

| BENCHMARK | TASK ID | SHIFT FRACTION ($\delta_k$) | SCALE FACTOR ($\alpha_k$) |
|---|---|---|---|
| **HIGH-DIMENSION** | SOURCE 1 | 0.025 | $-0.5$ |
| | SOURCE 2 | 0.05 | 0.5 |
| **LOW-DIMENSIONAL** | SOURCE 1 | 0.1 | 0.5 |
| | SOURCE 2 | 0.2 | $-0.5$ |
| | SOURCE 3 | 0.3 | 1.5 |

## F.2. Comprehensive Experimental Settings

We utilized the `OpenBox` library (Li et al., 2021; Jiang et al., 2024) to implement the baseline methods, which rely on `GPy` for Gaussian Process modeling. Our proposed RA-TBO framework is based on `PyTorch` and `GPyTorch` to leverage efficient GPU acceleration for training Deep Ensembles and GPs. All experiments were conducted on a workstation equipped with dual AMD EPYC 9654 96-Core Processors and one NVIDIA GeForce RTX 4090 GPU.

To ensure a robust and fair comparison, we adopt the default hyperparameter settings implemented in the `OpenBox` toolkit for all baseline methods. We use the `'random_scipy'` option, which combines a global random search with L-BFGS-B local optimization with gradients.

To adapt the offline dataset for the rank-based loss function, we employ a data augmentation strategy following the approach proposed in (Tan et al., 2025). Specifically, we construct fixed-length training lists by randomly sampling design-score pairs from the entire dataset for a specified number of iterations. The detailed hyperparameter settings for our RA-TBO framework are provided in Table 4.

*Table 4.* Detailed Hyperparameters for RA-TBO.

| Parameter | Value / Configuration |
| --- | --- |
| ***1. Target Task Surrogate (Gaussian Process)*** | |
| Kernel Structure | Matern Kernel ($\nu = 2.5$) with ARD and Constant Mean. |
| Likelihood | Gaussian Likelihood. |
| Optimizer | Adam optimizer with $lr = 0.01$. |
| Training Schedule | 500 iterations per cycle, with 3 restarts |
| ***2. Acquisition Function (RA-UCB)*** | |
| Exploration Weight ($\beta_t$) | Constant value set to 2. |
| Rank Entropy Weight ($\lambda_t$) | $\lambda_t = 0.2/\sqrt{t}$. |
| Anchor Points ($\mathcal{A}_{src}$) | constructed by selecting the Top-1 candidate from each source task ($M = 1$). |
| Optimization Method | Adam optimizer with $lr = 0.05$. |
| Optimization Budget | 80 steps with 5 random restarts. |
| ***3. Source Task Models (Deep Ensembles)*** | |
| Architecture | MLP with 2 hidden layers (128 units each) and GELU activation. |
| Ensemble Size | 5 independent models per task. |
| Precision | Automatic Mixed Precision (AMP) enabled. |
| ***4. Low-Dimensional Training Configs ($d \leq 6$)*** | |
| Value Model Training | Loss: MSE; Steps: 3000; Batch Size: 64; LR: $10^{-2}$; Weight Decay: $10^{-5}$. |
| Rank Model Training | Loss: Listnet; Steps: 500; Batch Size: 64; LR: $10^{-2}$; Weight Decay: $10^{-5}$. |
| List Generation | List size 80, generating 32 lists per training step. |
| ***5. High-Dimensional Training Configs ($d = 50$)*** | |
| Value Model Training | Loss: MSE; Steps: 3000; Batch Size: 256; lr: $10^{-3}$; Weight Decay: $10^{-4}$. |
| Rank Model Training | Loss: ListNet; Steps: 300; Batch Size: 256; lr: $10^{-3}$; Weight Decay: $10^{-4}$. |
| List Generation | List size 100, generating 128 lists per training step. |

### F.3. Additional Experiments

**Ablation experiments.** To verify the contribution of each component in the RA-TBO framework, we conducted ablation experiments on the Synthetic Benchmarks by comparing our full method (utilizing a value-based model and then quantifying the similarity between source and target tasks, the Listwise loss to construct rank-based models, and the RA-UCB acquisition function) against four variants: **w/o Rank Model (Value-based):** We replace the rank-based surrogate models for source tasks with standard value-based models trained via MSE loss.

**w/o Listwise Loss (Pairwise):** We replace the Listwise ranking loss with the Pairwise ranking loss to train the source surrogate models.

**w/o Rank Entropy (Standard UCB):** We remove the Rank Entropy term from the acquisition function and only utilize the standard UCB to guide the search.

**w/o Rank Similarity (Rank-based Model for Similarity):** We replace the value-based model with a rank-based model for quantifying the transfer similarity (then via Kendall's $\tau$) between the target and source tasks.

Figure 5 illustrates the optimization curves of the ablation experiments, which demonstrate that all variants fall short of the full RA-TBO framework. First, the performance gap between **w/o Rank Model** and the full method highlights the limitation of value-based transfer under distribution shifts. This validates that rank-based modeling is better suited to transfer learning due to its superior robustness, which effectively captures structural invariants. Second, employing a value-based model paired with Kendall's $\tau$ for similarity estimation maintains theoretical tractability while averting the premature loss of distributional details. Third, the superiority of the full method over **w/o Listwise Loss** verifies the advantage of the Listwise approach in constructing source surrogate models. This is because the Listwise objective, based on cross-entropy, prioritizes optimizing the global ranking distribution, whereas the Pairwise approach captures only local relative information. Last, the full framework converges faster than **w/o Rank Entropy**, validating that the REN term promotes efficient exploration by leveraging ordinal information. The seamless integration of these complementary components enables RA-TBO to achieve optimal performance.

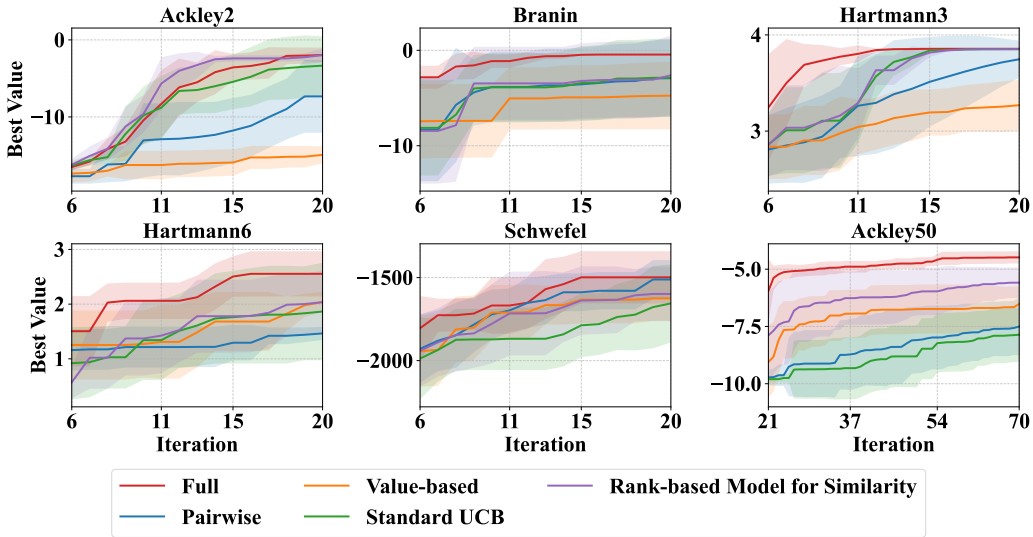

*Figure 5.* Optimization curves for ablation experiments.

**Computational complexity.** RA-TBO consists of offline and online phases. Given that the offline training of rank and value surrogate models is performed only once and can be precomputed during initialization of the target task, we analyze only the online phase.

We measured the wall-clock time per iteration across varying dimensionalities ($d$) by decomposing the total overhead into three distinct components: Target GP Construction, Kendall's $\tau$ Calculation, and Acquisition Function Optimization (including model normalization time via $\mathcal{X}_{norm}$ in Equation (5)). The analysis simulates a realistic transfer scenario with $K = 3$ source tasks (each containing $100d$ historical samples) and a current target dataset of $5d$ samples. The wall-clock time result is visualized in Figure 6. Benefiting from the GPU's parallel computing architecture, only GP training time exhibits a marked increase with dimensionality, whereas the increases for other operations are negligible. Even in the high-dimensional setting, the total online time remains on the order of seconds, which is insignificant compared to the black-box function's expensive, time-consuming evaluation cost.

**Sensitivity Analysis.** We further analyze the sensitivity of RA-TBO to two critical hyperparameters: the number of anchor points ($M \in \{1, \ldots, 10\}$) and the weight of the rank entropy term ($\lambda_0 \in [0.1, 1.0]$, where $\lambda_t = \frac{\lambda_0}{\sqrt{t}}$). We report the final optimization results (mean $\pm$ std) in Table 5. As observed, the performance fluctuations across varying settings of $M$ (Table 5a) and $\lambda_0$ (Table 5b) are not significant. This stability means that our method is robust to hyperparameter selection and does not require delicate tuning.

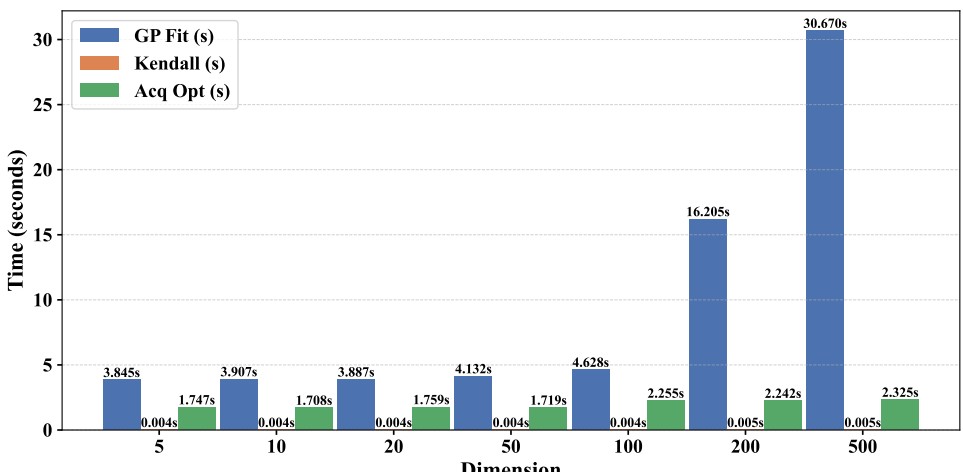

*Figure 6.* Wall-clock time analysis of the online optimization phase across varying dimensionalities.

*Table 5.* Sensitivity analysis of RA-TBO.

*(a)* Effect of Anchor Size ($M$)

| $M$ | ACKLEY2 | BRANIN | HARTMANN3 | HARTMANN6 | SCHWEFEL | ACKLEY50 |
|---|---|---|---|---|---|---|
| 1 | $-\mathbf{1.96} \pm \mathbf{0.97}$ | $-0.47 \pm 0.08$ | $\mathbf{3.86} \pm \mathbf{0.01}$ | $1.77 \pm 0.55$ | $-1745.82 \pm 214.67$ | $-\mathbf{4.43} \pm \mathbf{0.23}$ |
| 2 | $-2.10 \pm 1.35$ | $-0.95 \pm 0.71$ | $3.85 \pm 0.01$ | $2.17 \pm 0.59$ | $-1500.69 \pm 93.07$ | $-4.81 \pm 0.35$ |
| 3 | $-1.99 \pm 0.99$ | $-0.47 \pm 0.04$ | $3.84 \pm 0.02$ | $2.16 \pm 0.50$ | $-1694.65 \pm 218.12$ | $-4.96 \pm 0.59$ |
| 4 | $-2.07 \pm 0.86$ | $-0.47 \pm 0.05$ | $3.83 \pm 0.04$ | $2.14 \pm 0.56$ | $-\mathbf{1473.71} \pm \mathbf{85.11}$ | $-4.95 \pm 0.58$ |
| 5 | $-1.98 \pm 0.69$ | $-0.60 \pm 0.23$ | $3.85 \pm 0.02$ | $\mathbf{2.34} \pm \mathbf{0.65}$ | $-1593.42 \pm 96.20$ | $-5.60 \pm 1.36$ |
| 6 | $-2.96 \pm 0.19$ | $-\mathbf{0.46} \pm \mathbf{0.03}$ | $3.85 \pm 0.01$ | $2.07 \pm 0.55$ | $-1535.08 \pm 167.33$ | $-4.79 \pm 0.53$ |
| 7 | $-4.38 \pm 4.15$ | $-0.51 \pm 0.07$ | $3.85 \pm 0.01$ | $1.85 \pm 0.51$ | $-1474.38 \pm 52.87$ | $-4.82 \pm 0.50$ |
| 8 | $-3.86 \pm 4.99$ | $-0.49 \pm 0.06$ | $3.85 \pm 0.02$ | $1.73 \pm 0.53$ | $-1522.37 \pm 155.21$ | $-5.05 \pm 0.63$ |
| 9 | $-2.37 \pm 2.52$ | $-0.49 \pm 0.06$ | $3.85 \pm 0.01$ | $1.98 \pm 0.62$ | $-1655.50 \pm 356.06$ | $-5.14 \pm 0.39$ |
| 10 | $-4.42 \pm 4.70$ | $-0.47 \pm 0.04$ | $3.85 \pm 0.01$ | $2.18 \pm 0.64$ | $-1565.82 \pm 281.80$ | $-5.07 \pm 0.66$ |

*(b)* Effect of Rank Entropy Weight ($\lambda_0$)

| $\lambda_0$ | ACKLEY2 | BRANIN | HARTMANN3 | HARTMANN6 | SCHWEFEL | ACKLEY50 |
|---|---|---|---|---|---|---|
| 0.1 | $-2.27 \pm 1.40$ | $-0.94 \pm 0.72$ | $3.85 \pm 0.01$ | $2.06 \pm 0.67$ | $-1669.41 \pm 147.64$ | $-4.86 \pm 0.35$ |
| 0.2 | $-\mathbf{1.96} \pm \mathbf{0.97}$ | $-0.47 \pm 0.08$ | $\mathbf{3.86} \pm \mathbf{0.01}$ | $1.77 \pm 0.55$ | $-1745.82 \pm 214.67$ | $-\mathbf{4.43} \pm \mathbf{0.23}$ |
| 0.3 | $-2.58 \pm 0.83$ | $-0.90 \pm 0.59$ | $3.85 \pm 0.01$ | $1.94 \pm 0.55$ | $-1648.32 \pm 174.20$ | $-5.10 \pm 0.59$ |
| 0.4 | $-2.83 \pm 0.73$ | $-\mathbf{0.41} \pm \mathbf{0.01}$ | $3.85 \pm 0.01$ | $1.92 \pm 0.47$ | $-\mathbf{1486.01} \pm \mathbf{63.46}$ | $-5.04 \pm 0.61$ |
| 0.5 | $-2.75 \pm 2.49$ | $-0.44 \pm 0.02$ | $3.85 \pm 0.01$ | $2.08 \pm 0.53$ | $-1627.99 \pm 157.27$ | $-4.93 \pm 0.62$ |
| 0.6 | $-2.55 \pm 1.23$ | $-0.51 \pm 0.07$ | $3.85 \pm 0.01$ | $1.86 \pm 0.56$ | $-1556.04 \pm 149.31$ | $-4.97 \pm 0.39$ |
| 0.7 | $-2.96 \pm 0.78$ | $-0.63 \pm 0.13$ | $3.85 \pm 0.02$ | $1.98 \pm 0.57$ | $-1581.60 \pm 284.21$ | $-5.20 \pm 0.41$ |
| 0.8 | $-1.98 \pm 0.54$ | $-0.56 \pm 0.14$ | $3.85 \pm 0.01$ | $\mathbf{2.08} \pm \mathbf{0.59}$ | $-1640.10 \pm 282.15$ | $-5.08 \pm 0.50$ |
| 0.9 | $-3.47 \pm 2.19$ | $-0.54 \pm 0.14$ | $3.82 \pm 0.07$ | $1.98 \pm 0.52$ | $-1505.90 \pm 132.93$ | $-4.69 \pm 0.53$ |
| 1.0 | $-2.65 \pm 1.07$ | $-0.55 \pm 0.09$ | $3.85 \pm 0.02$ | $1.94 \pm 0.46$ | $-1557.25 \pm 123.29$ | $-4.89 \pm 0.58$ |

