# OpenReview forum: "RA-TBO: Rank-Aware Transfer Bayesian Optimization"
_ICML.cc/2026/Conference — Submitted to ICML 2026_

### Official Review · Reviewer_2tpN · 2026-03-06

**Soundness:** 3
**Presentation:** 2
**Significance:** 3
**Originality:** 3
**Overall Recommendation:** 4
**Confidence:** 3

**Summary:**

This paper proposes RA-TBO, a Transfer Bayesian Optimization framework that uses rank-based surrogate models for source tasks instead of value-based ones to achieve robust transfer under structural distortion and negative correlations between source and target tasks. The method decouples the transfer into an offline phase and an online phase. A novel Rank-Aware UCB acquisition function incorporating a Rank Entropy (REN) term is proposed to guide exploration. Theoretical results include concentration bounds for Kendall's τ estimation, optimization consistency, and no-regret guarantees. Experiments on synthetic benchmarks and real-world processor design space exploration (DSE) problems demonstrate improvements over existing TBO baselines.

**Compliance With Llm Reviewing Policy:**

Affirmed.

**Final Justification:**

W1 is resolved. The complexity analysis and empirical evidence (Table 5, Figure 6) are convincing.

W2 is largely addressed. The new DSE experiments with 14 source tasks and HPO experiments with 10 source tasks directly respond to our concern. The addition of MTGP and PFNs4BO baselines is appreciated. However, we note that on several HPO tasks, RA-TBO's margins over baselines are within standard deviation, suggesting the rank-based advantage is most pronounced under structural distortion/negative correlation rather than in general TBO settings. We encourage the authors to briefly discuss this applicability boundary in the revision.

I maintain my score at 4 (Weak Accept), which is already a positive recommendation for acceptance.

**Key Questions For Authors:**

See weakness

**Limitations:**

yes

**Strengths And Weaknesses:**

Strengths
-  The paper identifies an important and practical limitation of existing TBO methods, their inability to handle negative correlations and structural distortions between tasks. The motivating example in Figure 2 clearly illustrates why rank-based modeling is preferable to value-based modeling in this setting. Real-world DSE problems naturally exhibit such heterogeneity, making this a practically relevant contribution.
- The decoupled offline-online architecture is well-designed. Using rank-based loss functions for source tasks while retaining value-based GP for the target task is a sensible asymmetric design.
- The paper provides three meaningful theoretical results: (a) concentration bounds for Kendall's τ with both a general Hoeffding-type bound and an adaptive Bernstein-type bound that tightens when |τ_k| → 1; (b) Theorem 5.2 linking rank correlation to positional proximity of optima; (c) no-regret property of RA-UCB. The adaptive bound in Theorem 5.1 is particularly interesting as it shows faster convergence for highly correlated tasks.

Weaknesses
- The paper mentions computing covariance terms for the REN calculation jointly over candidate x and anchor set A_src, but does not provide a clear computational complexity analysis. For high-dimensional problems (d=50), the size of A_src (union of top-M points from K sources) and the cost of computing pairwise ranking probabilities could be significant.
- For experiment: (a) the synthetic benchmarks use only K=1 or K=2 source tasks, which is a rather limited multi-source setting. Real-world TBO scenarios often involve tens of source tasks.  (b) the real-world DSE experiments, while relevant, are limited to a specific domain (processor design). (c) The paper does not compare against some recent and relevant baselines such as methods that explicitly model task relationships (e.g., multi-task GP approaches with learned inter-task kernels) or more recent neural-network-based TBO methods.

---

> ### Author Rebuttal · Authors · 2026-03-30
>
> We appreciate your constructive comments and positive recognition of our motivation, framework design, and theoretical analysis.
>
> **W1: Lack of computational complexity for REN**
>
> **Response:**  Thank you for raising this important point. Below, we provide an explicit analysis of computational complexity of the REN and explain why the additional overhead remains acceptable even in high-dimensional settings.
>
> For a candidate $x$, REN evaluates its rank probabilities to the anchor set $\mathcal{A}\_{src}$. Let the joint query set size be $q = 1 + |\mathcal{A}\_{src}|$, and $n_t$ denote the number of target observations collected so far. Assuming the fusion model's covariance factorization is pre-computed, the REN computational cost per candidate is bounded by $T_{\mathrm{REN}}(x) = \mathcal{O}(n_t q^2 + q^3 + q)$. These three terms respectively account for constructing the target posterior covariance block, performing fused covariance matrix operations, and computing probabilities and entropies point by point. The anchor set size is strictly bounded by $|\mathcal{A}\_{src}| \le KM$, where $K$ is the number of source tasks and $M$ is the top samples per source. Notably, this additional REN cost depends mainly on $q$ and $n_t$.
> The main effect of dimensionality is on the overall optimization of the acquisition function over the search space, rather than on the REN-specific computation itself.
>
> As shown in Table 5, increasing $M$ does not significantly improve performance; therefore, $q$ remains small in practice. Meanwhile, with batched operations on GPUs, Figure 6 shows that the total wall-clock time for acquisition function optimization remains acceptable even in high-dimensional scenarios (even for $d=500$). The additional overhead of REN is modest compared to its benefits, as it improves sample efficiency and reduces the cost of expensive black-box evaluations.
>
> **W2: Concerns on experimental breadth and baseline coverage**
>
> **Response:**  Thank you for your helpful suggestion. We agree that our experiments can be strengthened by including more source tasks, a broader range of real-world applications, and additional baselines. To address this concern, we have made the following improvements to the paper：
>  - First, we newly introduced another CPU benchmark targeting 1-concurrent (**1-con**) and 8-concurrent (**8-con**) Shell Scripts as real-world workloads (subtasks in UnixBench), with **14** source tasks including 10 other core UnixBench metrics, 3 emuMySQL subtasks, and the Renaissance.
> - Second, we newly added hyperparameter optimization (via yahpo-gym) with more source tasks.
> - Third, we expanded our baselines to include a multi-task GP (**MTGP**) with learned inter-task kernels (implemented via BoTorch) and the recent neural-network-based TBO method PFNs4BO, as suggested by Reviewer kM1c in W2.
>
> The results of all additional experiments are summarized in the table below.  These additional experiments will be included in the revised paper.
>
> The new experiments follow the same setup as in the original paper. In the HPO problem, for each target task, we select **10** source tasks, each with 100 historical points, from different datasets in the same HPO benchmark family. All tasks optimize the same model class and metric under the same space, differing only in the dataset. For each target task, we start with 10 initial observations, perform 20 optimization iterations, and repeat with 5 random seeds.
>
> **Table: Performance comparison on additional real-world BO tasks**
> | Task and metric | SGPR | RGPE | TRANSBO | MTGP | PFNs4BO | OURS | OURS-PreferGP |
> |---|---:|---:|---:|---:|---:|---:|---:|
> | DSE(1-con)_improvement | 4.61%±0.45% | 4.59%±0.30% | 4.76%±0.49% | 4.38%±0.52% | 4.41%±0.33% | **5.13%±0.32%** | 4.75%±0.23% |
> | DSE(8-con)_improvement | 6.89%±0.28% | 6.45%±0.58% | 7.35%±0.56% | 5.92%±0.62% | 6.23%±0.77% | **8.32%±0.45%** | 7.92%±0.38% |
> | lcbench_acc | 0.7679±0.0083 | 0.7694±0.0053 | 0.7664±0.0107 | **0.7716±0.0012** | 0.7618±0.0006 | 0.7686±0.0056 | 0.7629±0.0121 |
> | lcbench_cross_entropy (minimize) | 0.4064±0.0344 | 0.4207±0.0437 | 0.4252±0.0432 | 0.3983±0.0082 | 0.3998±0.0330 | **0.3966±0.0213** | 0.3973±0.0323 |
> | rbv2_super_acc | 0.7978±0.0007 | 0.7981±0.0003 | 0.7966±0.0015 | 0.7977±0.0007 | 0.7972±0.0001 | **0.7983±0.0002** | 0.7982±0.0003 |
> | rbv2_super_bac | 0.7925±0.0264 | 0.7821±0.0299 | **0.8023±0.0033** | 0.7723±0.0382 | 0.7903±0.0216 | 0.7960±0.0125 | 0.7820±0.0231 |
> | svm_acc | 0.9337±0.0020 | 0.9342±0.0023 | 0.9304±0.0078 | 0.9306±0.0009 | **0.9356±0.0023** | 0.9346±0.0018 | 0.9325±0.0032 |
> | svm_bac | 0.5822±0.0943 | 0.6612±0.0019 | 0.6663±0.0572 | 0.6287±0.0411 | 0.6569±0.0479 | **0.6707±0.0219** | 0.6434±0.0327 |
> | xgboost_acc | 0.9322±0.0035 | 0.9335±0.0021 | 0.9314±0.0016 | 0.9291±0.0029 | 0.9332±0.0021 | **0.9339±0.0027** | 0.9335±0.0028 |
> | xgboost_bac | 0.7374±0.0057 | 0.7402±0.0076 | 0.7334±0.1378 | 0.7435±0.0103 | 0.7228±0.0493 | 0.7445±0.0028 | **0.7449±0.0035** |

---

> > ### Author Rebuttal · Reviewer_2tpN · 2026-04-03
> >
> > Thanks the authors for the thorough rebuttal.
> >
> > W1 is resolved. The complexity analysis and empirical evidence (Table 5, Figure 6) are convincing.
> >
> > W2 is largely addressed. The new DSE experiments with 14 source tasks and HPO experiments with 10 source tasks directly respond to our concern. The addition of MTGP and PFNs4BO baselines is appreciated. However, we note that on several HPO tasks, RA-TBO's margins over baselines are within standard deviation, suggesting the rank-based advantage is most pronounced under structural distortion/negative correlation rather than in general TBO settings. We encourage the authors to briefly discuss this applicability boundary in the revision.
> >
> > I maintain my score at 4 (Weak Accept), which is already a positive recommendation for acceptance.

---

> > > ### Author Response · Authors · 2026-04-04
> > >
> > > Thank you very much for your follow-up and for your positive recommendation.
> > >
> > > We sincerely appreciate your recognition that our rebuttal has adequately addressed W1 and largely addressed W2. We are also grateful for your constructive suggestion regarding the applicability boundary of RA-TBO. This is an important observation, and we will discuss this applicability boundary in the revised paper to better clarify the scope and strengths of our method.

---

### Official Review · Reviewer_kM1c · 2026-03-10

**Soundness:** 3
**Presentation:** 3
**Significance:** 2
**Originality:** 3
**Overall Recommendation:** 4
**Confidence:** 4

**Summary:**

This paper proposes RA-TBO, a transfer Bayesian optimization framework that improves robustness when source and target tasks have different scales or correlations. Instead of transferring absolute function values, it transfers ranking (ordinal) information from source tasks, which better captures shared global structure and avoids negative transfer. The method combines rank-based source models with a value-based target GP, weighting sources using Kendall’s rank correlation, and introduces a rank-aware UCB acquisition function for exploration. Experiments on synthetic and real optimization tasks show that RA-TBO achieves faster convergence and more reliable transfer than existing TBO methods.

**Compliance With Llm Reviewing Policy:**

Affirmed.

**Final Justification:**

The authors have addressed my main concerns, so I have decided to give this submission a positive score.

**Key Questions For Authors:**

- Is this method applicable to any surrogate model? If so, could the authors extend the method to PFNs?

**Limitations:**

Yes

**Strengths And Weaknesses:**

# Strengths

- The idea of using ranking-weighted function values, rather than absolute function values, to transfer knowledge from source tasks to the target domain is sound and intuitive.
- The presentation is clear and easy to follow, with helpful illustrations such as Figures 1 and 2 and Algorithm 1.
- The paper provides theoretical analysis of the proposed algorithm.

# Weaknesses

- The method appears to be limited to surrogate models based on deep ensembles. While GP models may not be suitable for conditioning on large-scale source data, other transfer mechanisms could be considered (e.g., transferring only the deep kernel).
- Recent approaches based on prior-data fitted networks (PFNs) [1, 2] are not compared. In particular, [3] proposes a PFN variant designed to handle distribution shift, which seems highly relevant to this work.
- The real-world experiments are somewhat limited. The paper mainly focuses on design space exploration (DSE) settings, whereas another important application of Bayesian optimization is hyperparameter optimization. It would strengthen the paper to demonstrate the effectiveness of the proposed method in additional BO applications.

### References

[1] Müller, Samuel, et al. "PFNs4BO: In-context learning for Bayesian optimization." International Conference on Machine Learning. PMLR, 2023.

[2] Hollmann, Noah, et al. "Accurate predictions on small data with a tabular foundation model." Nature 637.8045 (2025): 319–326.

[3] Helli, Kai, et al. "Drift-resilient TabPFN: In-context learning temporal distribution shifts on tabular data." Advances in Neural Information Processing Systems 37 (2024): 98742–98781.

---

> ### Author Rebuttal · Authors · 2026-03-30
>
> We sincerely thank you for your constructive feedback and for your appreciation of our core idea, framework, presentation, and theoretical analysis. Below, we provide our detailed responses.
>
> **W1: “The method appears to be limited to surrogate models based on deep ensembles”**
>
> **Response:** Thank you for your comment. We would like to clarify that the proposed RA-TBO framework is **model-agnostic** and is **not limited** to surrogate models based on deep ensembles. In this paper, we use deep ensembles only as the current implementation. More generally, the framework requires only a **value-based model** and a **rank-based** model.
>
> Additionally, we conducted new experiments without using deep ensembles to verify this point, as detailed in our response to Reviewer 2tpN in Q2.
> Specifically, we used a Preference GP as the rank-based model and standard GPs as the value-based models for both the source and target tasks (**OURS-PreferGP**). The corresponding results are provided in our response to Reviewer 2tpN in Q2. These results show that our framework remains effective under this alternative instantiation and still achieves strong performance.
>
> We will discuss this point more explicitly in the revised manuscript.
>
>
> **W2: Lack of comparison with PFN-based methods**
>
> **Response:**  We agree that PFNs represent an important and highly relevant recent research direction. However, we believe that our method and PFNs4BO are not directly comparable because they address fundamentally different problem settings:
> - Our framework focuses on transfer Bayesian optimization, where the goal is to accelerate optimization on a target task by leveraging historical data from correlated source tasks.
> - PFNs4BO is not designed for transfer Bayesian optimization. While it leverages synthetic priors via in-context learning to accelerate the optimization process, it does not transfer knowledge from task-specific historical datasets.
>
> Therefore, the two methods rely on different assumptions about the available information and are intended for different scenarios.
>
> In addition, [3] focuses on **temporal distribution shifts**, including Diabetes 130-US Hospitals (1999–2008), Electricity (weekly domains), and the Folktables US Census (yearly domains).
> By contrast, our setting concerns **source-target task transfer without temporal information**, such as transferring across different benchmarks for the same CPU (R-Line 15).
>
> Following your suggestion, we have included PFNs4BO as a new baseline in our experiments. The experimental results are provided in response to Reviewer 2tpN in Q2. PFNs4BO also performs very well. We will clarify these distinctions in the related work section and include these additional experiments.
>
> **W3: Broader real-world BO applications**
>
> **Response:**  Thank you for your very valuable suggestion. Following your suggestion, we conducted additional experiments on hyperparameter optimization (HPO) problems, and the results are shown in response to Reviewer 2tpN in Q2. The results show that our method also performs well on transferring HPO problems.
>
> **Q: Applicability to other surrogate models and PFNs**
>
> **Response:** Thank you for this question. As discussed in our response to W1, RA-TBO is a **model-agnostic** framework that requires only a value-based model and a rank-based model. It is therefore not restricted to any specific surrogate family. We have also verified this point empirically using a Preference GP (**OURS-PreferGP**) as the rank-based model, and the experimental results are provided in response to Reviewer 2tpN in Q2.
>
> In our framework, any surrogate model can be used as long as it satisfies the following requirements: the value-based model should provide estimates of the predictive mean and the joint distribution of the predictive objective values at two input points, while the rank-based model must additionally be capable of learning to rank.
>
> Extending RA-TBO to PFNs is possible in principle, but not as a direct plug-in. The key challenge is that, in our framework, the value-based model needs to output the covariance between the objective values at two input points. However, PFNs output the distribution for each input individually; therefore, they cannot be used directly as the value-based model in our framework unless additional techniques or assumptions are introduced to address this problem.
>
> Meanwhile, the rank-based model needs to learn to rank, which differs from the objective of PFNs. If we want to use PFN as the rank-based model in our framework, it would likely require nontrivial modifications to both the modeling and training procedures. We appreciate this insightful direction again. Exploring efficient adaptations to address PFNs remains a highly promising direction for our future work.

---

> > ### Author Rebuttal · Reviewer_kM1c · 2026-04-01
> >
> > Thanks for the rebuttal. It has addressed my concerns, and I will update my score to 4.

---

> > > ### Author Response · Authors · 2026-04-04
> > >
> > > Thank you very much for your positive feedback and for your support of our paper. We sincerely appreciate your careful reading and your recognition of our contribution. We will continue to improve the paper in the revision.

---

### Official Review · Reviewer_88S6 · 2026-03-12

**Soundness:** 2
**Presentation:** 3
**Significance:** 2
**Originality:** 3
**Overall Recommendation:** 3
**Confidence:** 2

**Summary:**

This paper studies transfer Bayesian optimization when source and target tasks may differ in scale, structure, or even correlation sign. To improve robustness, the authors propose RA-TBO, which transfers ranking information rather than relying mainly on absolute function values. The method estimates source-target similarity using expected Kendall’s tau, combines rank-based source models with a target GP surrogate, and further introduces a rank-aware acquisition function, RA-UCB. The paper also provides theoretical analysis and experiments on synthetic and real-world design space exploration tasks, showing competitive performance over several baselines.

**Compliance With Llm Reviewing Policy:**

Affirmed.

**Final Justification:**

I still have doubts about this work, so I will keep my score unchanged.

**Key Questions For Authors:**

- Q1: The paper emphasizes that RA-TBO can effectively exploit negatively correlated source tasks. However, the theoretical discussion seems to focus more on positively aligned rank consistency. Could the authors clarify whether there is a corresponding theoretical justification for the negative-correlation case, beyond the empirical results?

- Q2: The similarity estimation is based on expected Kendall’s tau computed over target observations, but BO samples are collected sequentially and adaptively rather than under an i.i.d. setting. How sensitive is the method and its analysis to this mismatch between theory and the actual BO process?

- Q3: In the synthetic experiments, many source tasks are generated through relatively controlled transformations such as scaling and shifting. Could the authors discuss how well RA-TBO is expected to generalize when source-target discrepancies are more complex and less structured than those considered here?

- Q4: The method uses value-based models for similarity estimation, but rank-based models for source knowledge transfer. What is the main reason for this design split, and is there evidence that directly using rank-based models for similarity estimation would be inferior?

**Limitations:**

yes

**Strengths And Weaknesses:**

## Strengths

- The paper addresses a relevant problem in transfer Bayesian optimization, especially when source and target tasks differ in scale, structure, or correlation.


- The core idea of transferring rank information rather than relying mainly on absolute function values is intuitive and reasonably novel. The overall framework is also technically coherent, including similarity estimation, surrogate fusion, and rank-aware acquisition.


- The empirical results are fairly strong on both synthetic benchmarks and DSE tasks, suggesting that the method is practically promising.


## Weaknesses

- The main claim about effectively exploiting negative correlations is stronger than what is directly supported by the theory, which appears to focus more on positively aligned rank consistency.


- Some theoretical assumptions seem idealized for sequential BO, where samples are collected adaptively rather than under a simple i.i.d. setting.

- The synthetic source-task construction is somewhat controlled and may be favorable to the proposed method, so robustness under more realistic cross-task discrepancies is still not fully clear.

---

> ### Author Rebuttal · Authors · 2026-03-30
>
> We sincerely thank you for your time and feedback. We address your concerns one by one below.
>
> **W1 and Q1: Lack of theory support for negative correlation**
>
> **Response:** Thank you for the question. Our framework, in fact, naturally accommodates the negative-correlation case through its symmetric mathematical design. Although Theorem 5.2 presents the positive-correlation case ($\tau\to1$) for illustration, the negative-correlation case follows directly by negating the source-task labels, which reverses the sign of Kendall’s $\tau_k$ in Eq. (3). Therefore, the case $\tau_k\to-1$ is mathematically equivalent to the case $\tau_k\to1$  after sign reversal.
>
> This symmetry is explicitly reflected in our method: we use $w_k=|\tau_k|$ in Eq. (4) and apply a sign correction $\text{sgn}(\tau_k)$ in the mean normalization term of Eq. (6). Accordingly, the proof only presents $\tau_k\to1$ as a representative case, since the negative-correlation case can be derived immediately through negation. We did briefly mention this logic in the original manuscript (R-Lines 191–194 and 211–215). We will clarify this point more clearly around Theorem 5.2 in the revised version.
>
> **W2 and Q2: Lack of modeling for sequential and adaptive BO process**
>
> **Response:** Thank you for raising this important point. In fact, we have already briefly discussed the sequential sampling setting in Appendix C.4 (Line 872), where we note that our results naturally extend to sequential BO through the Azuma–Hoeffding inequality in Martingale theory.
>
> In the BO process, adaptive sampling may be beneficial for optimization. As noted in Section 4.2 (L-Line 218), high-performing points gradually accumulate in the target dataset $\mathcal{D}_t$.  This naturally causes the similarity estimation to focus more on high-performance regions, which is well aligned with the maximization objective. The experimental results in Section 6 (Fig. 3 and Table 1) show that our method achieves good performance under sequential sampling.
>
> We will state this point more explicitly in the revised manuscript.
>
> **W3 and Q3: Lack of experimental validations of robustness under realistic cross-task discrepancies**
>
> **Response:** Thank you for raising this point. Our framework is designed to generalize to highly complex and unstructured cross-task discrepancies, and in fact, this has already been validated through extensive experiments on real-world datasets.
>
> Specifically, to evaluate generalization to unknown, unstructured problems, we tested our framework on 5 real-world DSE tasks as shown in Table 1.
>
> Additionally, we have newly added experiments on more real-world cases, including the UnixBench DSE problems and the hyperparameter optimization. Due to space limitations, please refer to our response to Reviewer 2tpN in Q2 for details. We will include these additional experiments in the revised manuscript.
>
> **Q4: What is the main reason for the design split between similarity estimation and knowledge transfer?**
>
> **Response:** Thank you for this question. The rationale behind this design split is discussed in Section 4.1 (L-Line 172). In addition, we provide supporting empirical evidence in an ablation study in Appendix F.3, which validates the effectiveness of separating similarity estimation from knowledge transfer.
>
> We briefly restate the reasons here for your reference:
>
> - For similarity estimation, we use value-based models because directly using rank-based models would prematurely discard predictive distribution details before projection into the rank space via Kendall’s $\tau$. In this way, the mechanism remains rank-based, and the fitted model can serve as the true source function when a sufficiently large dataset is available, as required by Theorem 5.1.
> - For knowledge transfer, we employ rank-based models because value-based methods always overfit absolute values and local outliers (Section 4.1).
>
> As additional empirical support, the ablation study (Fig. 5, Rank-based Model for Similarity) shows that directly using rank-based models for similarity estimation performs worse than our current design. We will clarify this motivation and the corresponding empirical evidence more explicitly in the revised manuscript.

---

> > ### Author Rebuttal · Reviewer_88S6 · 2026-04-03
> >
> > Thank you for the detailed rebuttal. Your response does help clarify some of my concerns and makes the design motivation of the method easier to understand. In particular, for Q4, the explanation of why value-based models are used for similarity estimation while rank-based models are used for knowledge transfer is much clearer now, and the supporting empirical evidence is helpful. I also appreciate the explanation for the negative-correlation case through sign reversal, absolute Kendall weighting, and sign correction in normalization, which makes the implementation logic more understandable.
> >
> > That said, I still feel that some of my main concerns are only partially addressed.
> >
> > - First, regarding the theoretical support for the negative-correlation case, the rebuttal mainly argues that this case can be reduced to the positive-correlation case through sign reversal, so the method is symmetric by design. This helps explain why the method can handle negative correlation in practice. However, my original question was more about whether there is a comparably explicit and formal theoretical justification for the negative-correlation case itself. At the moment, the response provides an intuitive equivalence argument rather than a fully stated parallel theoretical result. So while this point is clearer now, I do not think it is fully resolved.
> >
> > - Second, regarding the mismatch between the i.i.d. assumptions in the theory and the sequential, adaptive nature of BO, I still see this as a substantive issue. The rebuttal mentions that the sequential setting was briefly discussed in the appendix and that martingale/Azuma–Hoeffding arguments may extend the analysis. However, this currently reads more like a plausible direction for extension than a complete and verifiable treatment. **As a result, this response partly alleviates my concern, but it does not fully remove the gap between the analysis assumptions and the actual BO setting.**
> >
> > - Third, on the experimental side, I agree that the real-world DSE results provide some support for robustness under more complex source-target discrepancies. However, my original concern was that the synthetic source-task construction appears relatively controlled, mainly based on scaling and shifting transformations, which may still be somewhat favorable to the proposed method. The rebuttal mentions additional real-world experiments, which is helpful, but as presented here this still sounds more like material to be added in the revision rather than evidence already fully demonstrated in the current version. So I think the question of robustness under more complex and less structured discrepancies is still only partially addressed.
> >
> > Overall, the rebuttal improves the clarity of the paper and strengthens the motivation behind the method, but in my view several key points remain only partially resolved, especially the formal theoretical treatment of negative correlation, the theoretical connection to sequential BO, and the strength of the experimental evidence under less structured cross-task discrepancies. These aspects would benefit from clearer discussion and stronger support in the revision.

---

> > > ### Author Response · Authors · 2026-04-05
> > >
> > > We sincerely thank you for your constructive follow-up. We are glad to see that many of your concerns have been addressed, especially those regarding our core methodology.
> > >
> > > We greatly appreciate your acknowledgment (1) that the rationale behind using value-based models for similarity estimation and rank-based models for knowledge transfer is **now much clearer**, (2) that the negative-correlation handling through sign reversal, absolute Kendall weighting, and sign correction in normalization is **more understandable**, and (3) that the supporting DSE experiments **provide helpful empirical evidence** for the robustness of our framework.
> > >
> > > We would like to address your remaining three concerns one by one below.
> > >
> > > **1. Lack of an explicit and formal theoretical justification for the negative-correlation case**.
> > >
> > > **Response:** Thank you for this clarification. As you noted, our method is general due to its symmetric design, under which the negative-correlation case can be reduced to the positive-correlation case through sign reversal. Therefore, the negative-correlation case does not require separate treatment.
> > > After sign reversal, a negatively correlated source task becomes positively correlated under our similarity measure, and therefore **the negative-correlation case is naturally covered by the existing theoretical analysis**.
> > >
> > > We agree that this point should be stated more explicitly and formally, and that the theoretical value of explicitly incorporating the negative-correlation case should be better highlighted. We will revise the discussion around Theorem 5.2 to clarify this issue in the revised manuscript.
> > >
> > > **2. Lack of a theoretical analysis for non-iid sequential BO.**
> > >
> > > **Response:** Thank you for raising this important point. We agree that the current manuscript does not present a direct and self-contained theoretical analysis for the non-iid sequential BO setting.
> > >
> > > In fact, this issue was already considered in our original paper. Our current presentation first gives Theorem 5.1 for the basic setting, and then notes that the result can be extended to the sequential setting through Azuma-Hoeffding inequality, as described in Appendix C.4.
> > > We agree, however, that this presentation is not sufficiently explicit and that a complete and verifiable treatment would be much better.
> > >
> > > Following your suggestion, we have completed the corresponding theoretical analysis and plan to include it in the revised version. The main proof idea is as follows:  Since every pairwise comparison term is bounded, the corresponding martingale differences are also uniformly bounded. This allows a direct application of Azuma–Hoeffding, which yields an exponential concentration bound for the empirical Kendall coefficient around a sequential conditional mean. Therefore, although the sequential BO setting falls outside the standard i.i.d. framework, the concentration result can still be established through martingale arguments. Due to space limitations, we cannot include the full derivation here.
> > >
> > > **3. The synthetic source-task construction appears relatively controlled. Robustness under more complex and less structured discrepancies is still only partially addressed.**
> > >
> > > **Response:** Thank you. We appreciate your acknowledgment that the real-world DSE results provide supporting evidence for robustness under more complex source–target discrepancies.
> > >
> > > We would also like to clarify that **the synthetic experiments are intentionally designed as controlled settings** to simulate specific types of source–target discrepancy, such as shifting, scaling, and negative transfer, in order to validate the proposed method. Their purpose is therefore to demonstrate the feasibility and behavior of the method in a clean and interpretable setting.
> > >
> > > In contrast, we conduct extensive experiments on real-world datasets to evaluate the robustness of our method under more complex and less structured discrepancies. In total, we include **15** real-world experiments covering 2 major scenarios, including:
> > >
> > >  - DSE with Negative Correlations (in original paper). The experiment aims to transfer between Renaissance and emu-MySQL to demonstrate the algorithm's robustness against real-world, conflicting optimization objectives.
> > >  - DSE with Positive Correlations (in original paper). The experiment transfers knowledge among three emu-MySQL sub-benchmarks (Readonly, Writeonly, Default) to evaluate performance under implicit, positive structural relationships.
> > >  - **2** additional DSE tasks and **8** Hyperparameter Optimization tasks (newly added in rebuttal). The experiment aims to further validate the framework's practical robustness across completely diverse tasks where cross-task relationships are entirely unknown and inherently unstructured.
> > >
> > > Thank you again for this valuable suggestion. In the revised manuscript, we will make this distinction clearer in the experimental section and incorporate these additional experimental results.

---

### Official Review · Reviewer_GVNe · 2026-03-15

**Soundness:** 4
**Presentation:** 4
**Significance:** 4
**Originality:** 4
**Overall Recommendation:** 5
**Confidence:** 2

**Summary:**

This paper proposes a rank-based transfer BO method. The intuition is that, rank is more robust and reliable for transfer BO than the absolute value of the source dataset. Firstly, source's value-based model $V_k$ and rank-based model $F_k$ is learned respectively (with deep ensemble). and then $F_k$ is used to transfer, and $V_k$ is used to measure source-target similarity. Then, Kendall's $\tau$ is computed with the absolute value, which means that even negative correlation is useful for transfer, which is different from the previous transfer BO regime. Final transfer is done with gPoE fusion. Various theoretical analysis is provided, along with strong empirical performance than the previous value-based transfer BO methods.

**Compliance With Llm Reviewing Policy:**

Affirmed.

**Final Justification:**

Overall, I found the paper interesting, especially regarding its strong motivation and empirical results. I thus maintain my original score 5. However, I'm not expert on the theory part, and I understand there exists some issues raised by the authors. So I lower my confidence score accordingly. Nevertheless, except for the theory part, I strongly believe this submission will benefit the BO and HPO community in many aspects.

**Key Questions For Authors:**

Those questions are all minor.
- Maybe I missed something, but could you elaborate again when the rank-based method is superior to value-based and when not?
- The methodology, especially the transfer mechanism seems a little bit complicated. Is there any way to simplify the transfer method?
- What if we model target as rank (as well as source), rather than value?

**Limitations:**

The authors did not particularly addressed the limitations of this paper.

**Strengths And Weaknesses:**

**Strengths**
- **Motivation is really strong**. I personally have also thought of this problem as very important for many years. This submission seems to be a very successful and theoretically sound method among all I have seen so far. I strongly believe that the motivation of this paper is really strong and together with successful empirical results, this paper definitely deserves acceptance.
- Methodology seems sound, although I was not able to fully understand all of the details, honestly.
- Theoretical analysis is provided thoroughly, although some of the assumptions are a little bit strong. But that's understandable.
- Empirical performance is also really strong, outperforming all of the traditional transfer BO baselines by significant margin.

**Weaknesses**
- I don't find any specific weaknesses of this paper. That's maybe because for the methodology part, I wasn't able to understand all of the details. I will defer the discussion of the weaknesses of this paper in the discussion phase with other reviewers.

---

> ### Author Rebuttal · Authors · 2026-03-30
>
> We sincerely thank you for the strong support and the highly positive evaluation of our work. We are deeply encouraged by your appreciation of our motivation and theoretical analysis. We also appreciate your helpful feedback regarding the technical details of our methodology. Below, we provide detailed responses to your questions to further clarify the transfer mechanism.
>
> **Q1: When is rank-based modeling superior to value-based modeling?**
>
> **Response:** Thank you for this insightful comment.
> In summary, the choice between rank-based and value-based modeling should be understood as a trade-off between **information retention** and **robustness**.
> Value-based models preserve richer numerical information, whereas rank-based models are generally much more robust, particularly when transferring across heterogeneous metrics with different scales and magnitudes.
>
> - **When value-based is superior**:  Value-based modeling is more suitable when target data are limited and expensive. In such settings, preserving the full numerical information is important, as it enables more accurate estimation of both the predictive mean and uncertainty, both of which are crucial for downstream acquisition function design.
>
>
> - **When Rank-based is superior**: Rank-based modeling is more suitable when source tasks exhibit numerical deviations from the target task, while still preserving a broadly consistent ordering structure. By focusing on relative ordering rather than absolute values, it is substantially more robust and naturally mitigates heterogeneity in scale and magnitude.
>
> We will provide a more systematic discussion of this problem in the revised manuscript.
>
> **Q2: Simplifying the Transfer Method**
>
> **Response:** Thank you for this helpful suggestion. We will provide a clearer, step-by-step explanation of the transfer mechanism in the revised manuscript.
>
> Specifically, our framework consists of two modules, an offline module and an online module, each serving a distinct and interpretable role, as shown in Fig. 1 of our paper.
>
> - **Offline:** We train both a rank-based model and a value-based model using the source data.
>
> - **Online:** The value-based model is used to quantify task similarity, while the rank-based model transfers source knowledge and dynamically integrates it with a Gaussian process trained on the target data. The resulting fused surrogate model is then used by a rank-aware UCB criterion to select the next query.
>
> As shown in the ablation study in Appendix F.3, removing any of these components leads to a clear degradation in transfer performance. In the revision, we will further refine Fig. 1 with clearer annotations and improve the presentation of the full pipeline to make the method easier to follow.
>
> **Q3: "What if we model target as rank (as well as source), rather than value?"**
>
> **Response:** Thank you for this insightful question, which is closely related to the discussion in Q1. While modeling the target task in rank space may appear more structurally consistent with the source-side design, it would discard valuable numerical information contained in the costly target evaluations. If we model the target with a rank-based approach, some important information may be lost, reducing optimization efficiency.
>
> **Limitation: Lack of discussion of limitations**
>
> **Response:** Thank you for pointing this out. Our method has the following two limitations that we will make more explicit in the revised manuscript.
>
> First, the method relies on sufficiently rich source-task data to support effective transfer.
> Second, as clarified in our response to Reviewer kM1c in W1 and the Questions, the value-based model is required to reliably estimate both the predictive mean and the joint distribution of predictive objective values at two input points, while the rank-based model must additionally be capable of learning to rank.
>
> These two limitations may limit the applicability of our method when source-task data are limited or when candidate models require nontrivial adaptation to meet these requirements.

---

> > ### Author Rebuttal · Reviewer_GVNe · 2026-03-31
> >
> > Thanks for the rebuttal. I don't have any other specific questions or comments. I maintain my rating accordingly.

---

> > > ### Author Response · Authors · 2026-04-04
> > >
> > > Thank you again for your high evaluation of our work, especially for your recognition of its motivation and theoretical analysis.

---

### Decision · Program_Chairs · 2026-04-30

**Decision:**

Reject

**Comment:**

This paper tackles Transfer Bayesian Optimization (TBO) in settings where source and target tasks exhibit structural distortions or negative correlations. While reviewers agreed that the problem of handling structural distortions and negative correlations in Transfer Bayesian Optimization (TBO) is highly relevant, and that the core idea of transferring rank information rather than absolute values is intuitive, they also raised valid and important concerns about the paper's empirical completeness, theoretical soundness, and relationship to prior literature.

Specifically, the main issues are:
* **Shortcomings of the Empirical Evaluation:** The original submission relied entirely on synthetic functions and a highly specific domain (processor Design Space Exploration). For a BO paper, omitting standard HPO benchmarks in the initial submission is a significant gap. Furthermore, the main DSE experiments report only a single final normalized improvement percentage, thus not providing any information about "anytime performance", which is critical for BO approaches.
* **Lack of Statistical Significance in General Settings:** While the authors provided a substantial amount of new HPO data during the rebuttal period to address the missing baselines, these late additions bypassed the standard peer-review scrutiny. More critically, as noted in the discussion, this new data revealed that RA-TBO's performance margins over the baselines on several of the newly added HPO tasks fall within the standard deviation. This suggests the benefit of the method might be limited specifically to cases of severe structural distortion rather than general TBO settings. A more careful and comprehensive analysis is required.
* **Theoretical Gaps and "Promissory Notes"**: The paper claims to handle negative correlations, but the core theory explicitly focuses on positively aligned rank consistency. While the authors provided outlines for new proofs (e.g., utilizing the Azuma-Hoeffding inequality) during the rebuttal, a paper at this venue must be evaluated on the proofs it contains, not the promise of proofs to be added in the camera-ready revision.
* **Gap in Discussion of Related Work:** The core contribution of this paper involves converting absolute numerical source data into pairwise probabilities to bypass scale distortion, which is then used to guide the target optimization process. This mechanism is closely related to the rich literature on Preferential Bayesian Optimization (PBO). PBO explicitly deals with utilizing pairwise preference judgments to learn latent utility functions, particularly when absolute numerical evaluations are unreliable, distorted, or unavailable. While this was not flagged in reviews and discussion, I believe this is a nontrivial omission, and the relationship between the proposed approach and PBO should be discussed in the paper.

Therefore, the paper cannot be accepted in its current form.

However, the work certainly has potential, and I encourage the authors to address the identified gaps in a revised version of the paper and resubmit to an appropriate future venue.